# STOCHASTIC ADAPTIVE SEQUENTIAL BLACK-BOX OPTIMIZATION FOR DIFFUSION TARGETED GENERATION

## ABSTRACT

Diffusion models have demonstrated great potential in generating high-quality content for images, natural language, protein domains, etc. However, how to perform user-preferred targeted generation via diffusion models with only black-box target scores of users remains challenging. To address this issue, we first formulate the fine-tuning of the inference phase of a pre-trained diffusion model as a sequential black-box optimization problem. Furthermore, we propose a novel stochastic adaptive sequential optimization algorithm to optimize cumulative black-box scores under unknown transition dynamics. Theoretically, we prove a $O(\frac{d^2}{\sqrt{T}})$ convergence rate for cumulative convex functions without smooth and strongly convex assumptions. Empirically, we can naturally apply our algorithm for diffusion black-box targeted generation. Experimental results demonstrate the ability of our method to generate target-guided images with high target scores.

## 1 INTRODUCTION

Diffusion models have shown great success in generating high-quality content in various domains, such as image generation (Rombach et al., 2022; Ramesh et al., 2022), video generation (Ho et al., 2022), speech generation (Kim et al., 2022b; Kong et al., 2020), and natural language generation (Hu et al., 2023; He et al., 2023). Thanks to the super-promising power of the diffusion models, guided sampling via diffusion models to achieve desired properties recently emerged and shown fantastic potential in many applications, e.g., text-to-image generation (Kim et al., 2022a), image-to-image translation (Tumanyan et al., 2023), protein design (Lee et al., 2023; Gruver et al., 2023).

Despite the popularity and success of diffusion models, how to employ diffusion models to generate user-preferred content with black-box target scores while avoiding re-training from scratch is still challenging and unexplored. One direct idea is to treat this problem as a black-box optimization problem and employ black-box optimization techniques (Audet & Hare, 2017; Alarie et al., 2021; Doerr & Neumann, 2019) to perform the fine-tuning of a pre-trained diffusion model with only black-box target scores. However, naively applying black-box optimization methods to optimize diffusion model parameters faces high-dimensional optimization challenges, which are prohibitive to achieving a meaningful solution in a feasible time.

More importantly, current black-box optimization techniques, e.g., Bayesian optimization techniques (Srinivas et al., 2010; Gardner et al., 2017; Nayebi et al., 2019), Evolution strategies (ES) or stochastic zeroth-order optimization (Back et al., 1991; Hansen, 2006; Wierstra et al., 2014; Lyu & Tsang, 2021; Liu et al., 2018; Wang et al., 2018) and genetic algorithms (Srinivas & Patnaik, 1994; Mirjalili & Mirjalili, 2019), are designed for single objective without considering the dynamic nature of sequential functions. As a result, we can not directly apply them to diffusion models due to ignoring the sequential nature of the generation process of diffusion models.

In this paper, we dig into the transition dynamic of the inference of diffusion models. By leveraging the relationship between the inference of diffusion models and the Stochastic Differential Equation (SDE) solver (Song et al., 2020; Lu et al., 2022a), we naturally formulate the fine-tuning of the inference of diffusion models as a black-box sequential optimization problem.

To solve the black-box sequential optimization problem, we propose a novel stochastic adaptive black-box sequential optimization algorithm by explicitly handling the history trajectory dependency in the cumulative black-box target functions. Our method performs full covariance matrix adaptive

updates that can take advantage of second-order information to deal with ill-conditioned problems. Theoretically, we prove a $O(\frac{d^2}{\sqrt{T}})$ convergence rate for convex functions without smooth and strongly convex assumptions. Thus, our method can handle non-smooth problems. Our contributions are listed as follows:

- We formulate the fine-tuning of the inference of the diffusion model as a black-box sequential optimization problem for black-box diffusion targeted generation.
- We proposed a novel stochastic adaptive black-box sequential optimization (SABSO) algorithm. Our SABSO can perform a full covariance matrix update to exploit the second-order information. Theoretically, we prove a $O(\frac{d^2}{\sqrt{T}})$ convergence rate for convex functions without smooth and strongly convex assumptions. Thus, our theoretical analysis can handle non-smooth problems. The convergence analysis of full matrix adaptive black-box optimization for convex functions without the smooth and strongly convex assumptions is technically challenging. Technically, we add a $\gamma_t$ enlargement term in the gradient update. This technique enables us to construct feasible solution sets of the adaptive update matrix during the whole algorithm running process to ensure convergence. To the best of our knowledge, our SABSO algorithm is the first full covariance matrix adaptive black-box optimization method that achieves a provable $O(\frac{d^2}{\sqrt{T}})$ convergence rate for convex functions without smooth and strongly convex assumptions.
- Empirically, we can naturally apply our algorithm for diffusion black-box targeted generation. Experimental results demonstrate the ability of our method to generate target-guided images with high black-box target scores.

## 2 PRELIMINARY BACKGROUND

### 2.1 DIFFUSION MODEL SAMPLING

The sampling phase of the diffusion model from noise to image can be implemented by solving the stochastic differential equation (SDE) (Song et al., 2020; Kingma et al., 2021) as in Eq.(1):

$$d\boldsymbol{x}_s = [\hat{f}(s)\boldsymbol{x}_s + \frac{g(s)^2}{\sigma_s}\boldsymbol{\epsilon}_\phi(\boldsymbol{x}_s, s)]\mathrm{d}s + g(s)\mathrm{d}\bar{\boldsymbol{w}}_s \tag{1}$$

where $\bar{\boldsymbol{w}}_s$ is the reverse-time Wiener process, $s$ denotes time changing from $K$ to 0, $\boldsymbol{\epsilon}_\phi(\boldsymbol{x}_s, s)$ denotes the diffusion model noise prediction with input $\boldsymbol{x}_s$ and time $s$. And $\sigma_s$ denotes the standard deviation of the diffusion noise scheme at time $s$. And $\hat{f}(s) := \frac{\mathrm{d}\log\alpha_s}{\mathrm{d}s}$, where $\alpha_s$ denotes the scaling parameter scheme in the diffusion model. And $g(s)^2 := \frac{\mathrm{d}\sigma_s^2}{\mathrm{d}s} - 2\hat{f}(s)\sigma_s^2$ (Kingma et al., 2021).

Recently, Lu et al. (2022a;b) proposed a DPM solver for solving the diffusion SDE with a small number of samples. The first-order SDE DPM solver is given in Eq.(2).

$$\boldsymbol{x}_s = \frac{\alpha_s}{\alpha_{s'}}\boldsymbol{x}_{s'} - 2\sigma_s(e^h - 1)\boldsymbol{\epsilon}_\phi(\boldsymbol{x}_{s'}, s') + \sigma_s\sqrt{e^{2h} - 1}\boldsymbol{z} \tag{2}$$

where $\boldsymbol{z} \sim \mathcal{N}(\boldsymbol{0}, \boldsymbol{I})$, $h = \lambda_s - \lambda_{s'}$, and $\lambda_s = \log(\alpha_s/\sigma_s)$ and $\lambda_{s'} = \log(\alpha_{s'}/\sigma_{s'})$. And $\alpha_s, \alpha_{s'}$ denote the scaling parameter at step $s$ and $s'$ in diffusion model, respectively.

### 2.2 BLACK-BOX OPTIMIZATION

Given a proper function $f(\boldsymbol{x}) : \mathbb{R}^d \to \mathbb{R}$ such that $f(\boldsymbol{x}) > -\infty$, black-box optimization is to minimize $f(\boldsymbol{x})$ by using function queries only. Instead of optimizing the original problem directly, ES or stochastic zeroth-order optimization methods optimize a relaxation of the problem $J(\theta) := \mathbb{E}_{p(\boldsymbol{x};\theta)}[f(\boldsymbol{x})]$ w.r.t. the parameter $\theta$ of the sampling distribution of the relaxed problem.

Evolution strategies (Rechenberg & Eigen, 1973; Nesterov & Spokoiny, 2017; Liu et al., 2018) employ a Gaussian distribution $\mathcal{N}(\boldsymbol{\mu}, \sigma^2\boldsymbol{I})$ with a fixed variance for candidate sampling. The approximate gradient descent update is given as

$$\boldsymbol{\mu}_{t+1} = \boldsymbol{\mu}_t - \frac{\beta}{N\sigma}\sum_{i=1}^{N}\boldsymbol{\epsilon}_i f(\boldsymbol{\mu}_t + \sigma\boldsymbol{\epsilon}_i), \tag{3}$$

where $\boldsymbol{\epsilon}_i \sim \mathcal{N}(\mathbf{0}, \boldsymbol{I})$ and $\beta$ denotes the step-size, and $\boldsymbol{\mu}_t$ denotes the mean parameter of the Gaussian distribution for candidate sampling at $t^{th}$ black-box optimization iteration.

The ES methods only perform the first-order approximate gradient update, the convergence speed is limited. Wierstra et al. (2014) proposed the natural evolution strategies (NES), which perform the approximate natural gradient update, in which A Gaussian distribution $\mathcal{N}(\boldsymbol{\mu}, \boldsymbol{\Sigma})$ is employed for sampling. Besides the updating of parameter $\boldsymbol{\mu}$, the covariance matrix $\boldsymbol{\Sigma}$) is also updated. Lyu & Tsang (2021) proposed an implicit natural gradient optimizer (INGO) for black-box optimization, which provides an alternative way to compute the natural gradient update. In INGO update rule, the inverse covariance matrix $\boldsymbol{\Sigma}^{-1}$ is updated instead of the covariance matrix $\boldsymbol{\Sigma}$. Moreover, CMAES (Hansen, 2006) provides a more sophisticated update rule and performs well on a wide range of black-box optimization problems. Despite the success of these methods, all these methods ignore the dynamic nature of the target function.

The recent work (Krishnamoorthy et al., 2023) introduces Denoising Diffusion Optimization Models (DDOM) for solving offline black-box optimization tasks using diffusion models. This method can also be naturally extended to black-box targeted generation tasks. The DDOM relies on an offline conditional model trained with reweighted data sampling. The generation is performed conditioned on a high target score. The pre-collected data set has a crucial influence on DDOM generation.

## 3 NOTATION AND SYMBOLS

Denote $\| \cdot \|_2$ and $\| \cdot \|_F$ as the spectral norm and Frobenius norm for matrices, respectively. Define $\text{tr}(\cdot)$ as the trace operation for matrix. Notation $\| \cdot \|_2$ will also denote $l_2$-norm for vectors. Symbol $\langle \cdot, \cdot \rangle$ denotes inner product under $l_2$-norm for vectors and inner product under Frobenius norm for matrices. For a positive semi-definite matrix $C$, define $\|\boldsymbol{x}\|_C := \sqrt{\langle \boldsymbol{x}, C\boldsymbol{x} \rangle}$. Denote $\mathcal{S}^+$ and as the set of positive semi-definite matrices. Denote $\Sigma^{\frac{1}{2}}$ as the symmetric positive semi-definite matrix such that $\Sigma = \Sigma^{\frac{1}{2}}\Sigma^{\frac{1}{2}}$ for $\Sigma \in \mathcal{S}^+$. Denote $\bar{\boldsymbol{x}}_k = [\boldsymbol{x}_1^\top, \cdots, \boldsymbol{x}_k^\top]^\top \in \mathcal{R}^{kd}$, where $\boldsymbol{x}_i \in \mathcal{R}^d$, $d$ denotes the dimension of the data. Denote $\bar{\boldsymbol{\mu}}_k = [\boldsymbol{\mu}_1^\top, \cdots, \boldsymbol{\mu}_k^\top]^\top$ and $\bar{\boldsymbol{\Sigma}}_k = diag(\boldsymbol{\Sigma}_1, \cdots, \boldsymbol{\Sigma}_k) \in \mathcal{R}^{kd \times kd}$ as the mean and diagonal block-wise covariance matrix for Gaussian distribution, respectively. Denote $\bar{\theta}_k := \{\bar{\boldsymbol{\mu}}_k, \bar{\boldsymbol{\Sigma}}_k\}$ as the parameter of the distribution for candidate sampling in black-box optimization and $\theta_k := \{\boldsymbol{\mu}_k, \boldsymbol{\Sigma}_k\}$ as its $k^{th}$ component. Denote $\bar{\theta}_k^t := \{\bar{\boldsymbol{\mu}}_k^t, \bar{\boldsymbol{\Sigma}}_k^t\}$ as the parameter at $t^{th}$ iteration.

## 4 METHOD

### 4.1 BLACK-BOX OPTIMIZATION FOR DIFFUSION TARGETED GENERATION

Diffusion model sampling can be implemented by solving Diffusion SDEs (Song et al., 2020). From SDE solvers in Eq.(2) , the inference phase of the diffusion SDE model can be rewritten as

$$\boldsymbol{x}_k = \widehat{\boldsymbol{\mu}}_\phi(\boldsymbol{x}_{k-1}, k-1) + \tilde{\sigma}_k \mathcal{N}(\mathbf{0}, \boldsymbol{I}) \tag{4}$$

for $k \in \{1, \cdots, K\}$, $k = K - s$, and $\boldsymbol{x}_0 \sim \mathcal{N}(\mathbf{0}, \boldsymbol{I})$, $\tilde{\sigma}_k$ denotes the coefficient of the SDE solver. And $\widehat{\boldsymbol{\mu}}_\phi(\boldsymbol{x}_{k-1}, k-1)$ is the prediction of a pre-trained diffusion model with fixed parameter $\phi$. The concrete $\widehat{\boldsymbol{\mu}}_\phi(\boldsymbol{x}_{k-1}, k-1)$ depends on the solver type.

To perform guided sampling from the diffusion model with the black-box target score function $F(\boldsymbol{x})$ (e.g., CLIP model evaluates on the input image $\boldsymbol{x}$), we generalize the inference of SDE solver as

$$\boldsymbol{x}_k = \widehat{\boldsymbol{\mu}}_\phi(\boldsymbol{x}_{k-1}, k-1) + \tilde{\sigma}_k \mathcal{N}(\boldsymbol{\mu}_k, \boldsymbol{\Sigma}_k) \tag{5}$$

for $k \in \{1, \cdots, K\}$, and fine-tuning the parameter $\theta_k := \{\boldsymbol{\mu}_k, \boldsymbol{\Sigma}_k\}$ for $k \in \{1, \cdots, K\}$. This naturally leads to a black-box optimization of the cumulative target score as

$$\tilde{J}(\bar{\theta}_K) := \mathbb{E}_{\boldsymbol{x}_{0:K} \sim p_{\bar{\theta}_K}} \Big[ \sum_{k=1}^{K} F(\boldsymbol{x}_k) \Big] \tag{6}$$

where $\boldsymbol{x}_k$ transition as $\boldsymbol{x}_k = \widehat{\boldsymbol{\mu}}_\phi(\boldsymbol{x}_{k-1}, k-1) + \tilde{\sigma}_k \boldsymbol{\epsilon}_k$ with $\boldsymbol{\epsilon}_k \sim \mathcal{N}(\boldsymbol{\mu}_k, \boldsymbol{\Sigma}_k)$, and $\bar{\theta}_K := \{\bar{\boldsymbol{\mu}}_K, \bar{\boldsymbol{\Sigma}}_K\}$, $F(\boldsymbol{x}_k)$ denotes the score function evaluated on input data $\boldsymbol{x}_k$. It is worth to remark that $\boldsymbol{x}_k$ depends on the output state $\boldsymbol{x}_{k-1}$. Problem (6) is essentially a sequential optimization problem. The strategy of choosing $\theta_k := \{\boldsymbol{\mu}_k, \boldsymbol{\Sigma}_k\}$ depends on the strategy of choosing $\{\theta_{k-1}, \theta_{k-2}, \cdots, \theta_1\}$ in a nested manner.

Instead of directly optimizing the objective Eq.(6), we optimize an augmented objective Eq.(7) in the noise space.

$$J(\bar{\theta}_K) := \mathbb{E}_{\boldsymbol{x}_0 \sim \mathcal{N}(\boldsymbol{0},\boldsymbol{I})} \mathbb{E}_{\bar{\epsilon}_K \sim \mathcal{N}(\bar{\boldsymbol{\mu}}_K, \bar{\boldsymbol{\Sigma}}_K)} \big[ \sum_{k=1}^{K} f_k(\bar{\epsilon}_k) \big] \tag{7}$$

where $\bar{\epsilon}_k = [\epsilon_1^\top, \cdots, \epsilon_k^\top]^\top$ for $k \in \{1, \cdots, K\}$, and $f_k(\bar{\epsilon}_k) = F_k(\boldsymbol{x}_k)$ for $\boldsymbol{x}_k$ transitioned as $\boldsymbol{x}_k = \hat{\boldsymbol{\mu}}_\phi(\boldsymbol{x}_{k-1}, k-1) + \tilde{\sigma}_k \epsilon_k$ with the trajectory $[\epsilon_1, \cdots, \epsilon_k]$. $F_k(\boldsymbol{x}_k)$ performs a deterministic sampling process $\boldsymbol{x}_{k+1} = \hat{\boldsymbol{\mu}}_\phi(\boldsymbol{x}_k, k) + \tilde{\sigma}_{k+1} \boldsymbol{\mu}_{k+1}$ for the future steps to achieve a predicted $\boldsymbol{x}_K$ and evaluate on the predicted $\boldsymbol{x}_K$.

The objective Eq.(7) can be rewritten as

$$J(\bar{\theta}_K) = \sum_{k=1}^{K} J_k(\bar{\theta}_k) = \sum_{k=1}^{K} \mathbb{E}_{\boldsymbol{x}_0 \sim \mathcal{N}(\boldsymbol{0},\boldsymbol{I})} \mathbb{E}_{\bar{\epsilon}_k \sim \mathcal{N}(\bar{\boldsymbol{\mu}}_k, \bar{\boldsymbol{\Sigma}}_k)} [f_k(\bar{\epsilon}_k)] \tag{8}$$

where $J_k(\bar{\theta}_k) = \mathbb{E}_{\boldsymbol{x}_0 \sim \mathcal{N}(\boldsymbol{0},\boldsymbol{I})} \mathbb{E}_{\bar{\epsilon}_k \sim \mathcal{N}(\bar{\boldsymbol{\mu}}_k, \bar{\boldsymbol{\Sigma}}_k)} [f_k(\bar{\epsilon}_k)]$.

## 4.2 CLOSED-FORM UPDATE RULE

In this section, we derive the update rule of the parameter to optimize Eq.(7). Without loss of generality, we assume minimization in this paper.

Given a parameter $\bar{\theta}_K^t$ at $t^{th}$ iteration, we aim to find a new parameter $\bar{\theta}_K^{t+1}$ by minimizing the objective difference as

$$\min_{\bar{\theta}_K} J(\bar{\theta}_K) - J(\bar{\theta}_K^t) \tag{9}$$

However, it is challenging to solve the optimization (9) directly. We thus optimize an approximation by first order Taylor expansion. We add a KL-divergence regularization to ensure $q_{\bar{\theta}_K}$ and $q_{\bar{\theta}_K^t}$ close, thus to keep the approximation accurate. The new optimization problem is given as

$$\min_{\bar{\theta}_K} J(\bar{\theta}_K) - J(\bar{\theta}_K^t) + \mathrm{KL}(q_{\bar{\theta}_K} | q_{\bar{\theta}_K^t}) = \sum_{k=1}^{K} J_k(\bar{\theta}_k) - J_k(\bar{\theta}_k^t) + \mathrm{KL}(q_{\bar{\theta}_K} | q_{\bar{\theta}_K^t}) \tag{10}$$

$$\approx \sum_{k=1}^{K} \left\langle \bar{\theta}_k - \bar{\theta}_k^t, \beta_k \nabla_{\bar{\theta}_k^t} J_k(\bar{\theta}_k^t) \right\rangle + \mathrm{KL}(q_{\bar{\theta}_K} | q_{\bar{\theta}_K^t}) \tag{11}$$

where $q_{\bar{\theta}_K} := \mathcal{N}(\bar{\boldsymbol{\mu}}_K, \bar{\boldsymbol{\Sigma}}_K)$ and $q_{\bar{\theta}_K^t} := \mathcal{N}(\bar{\boldsymbol{\mu}}_K^t, \bar{\boldsymbol{\Sigma}}_K^t)$.

Note that the problem (11) is convex w.r.t. $\bar{\theta}_K := \{\bar{\boldsymbol{\mu}}_K, \bar{\boldsymbol{\Sigma}}_K\}$, by setting the derivative to zero, we can achieve a closed-form update as

$$\boldsymbol{\mu}_k^{t+1} = \boldsymbol{\mu}_k^t - \sum_{i=k}^{K} \beta_i \mathbb{E}_{\boldsymbol{x}_0 \sim \mathcal{N}(\boldsymbol{0},\boldsymbol{I})} \mathbb{E}_{\mathcal{N}(\bar{\boldsymbol{\mu}}_i, \bar{\boldsymbol{\Sigma}}_i)} [(\epsilon_i - \boldsymbol{\mu}_i^t) f_i(\bar{\epsilon}_i)] \tag{12}$$

$$\boldsymbol{\Sigma}_k^{t+1^{-1}} = \boldsymbol{\Sigma}_k^{t^{-1}} + \sum_{i=k}^{K} \beta_i \mathbb{E}_{\boldsymbol{x}_0 \sim \mathcal{N}(\boldsymbol{0},\boldsymbol{I})} \mathbb{E}_{\mathcal{N}(\bar{\boldsymbol{\mu}}_i, \bar{\boldsymbol{\Sigma}}_i)} [(\boldsymbol{\Sigma}_i^{t^{-1}} (\epsilon_i - \boldsymbol{\mu}_i^t)(\epsilon_i - \boldsymbol{\mu}_i^t)^\top \boldsymbol{\Sigma}_i^{t^{-1}} - \boldsymbol{\Sigma}_i^{t^{-1}}) f_i(\bar{\epsilon}_i)] \tag{13}$$

Detailed derivation can be found in Appendix B.

To compute the update in Eq.(12) and Eq.(13), we perform Monte Carlo sampling by taking $N$ i.i.d. sequence $\{\boldsymbol{x}_0^j, \epsilon_1^j, \cdots, \epsilon_K^j\}$ for $j \in \{1, \cdots, N\}$, where $\boldsymbol{x}_0^j \sim \mathcal{N}(\boldsymbol{0}, \boldsymbol{I})$ and $\epsilon_k^j \sim \mathcal{N}(\boldsymbol{\mu}_k^t, \boldsymbol{\Sigma}_k^t)$ for $k \in \{1, \cdots, K\}$. This leads to unbiased estimators of the RHS of Eq.(12) and Eq.(13) as follows:

$$\boldsymbol{\mu}_k^{t+1} = \boldsymbol{\mu}_k^t - \frac{1}{N} \sum_{i=k}^{K} \beta_i \sum_{j=1}^{N} [(\epsilon_i^j - \boldsymbol{\mu}_i^t)(f_i(\bar{\epsilon}_i^j) - f_i(\bar{\boldsymbol{\mu}}_i^t))] \tag{14}$$

$$\boldsymbol{\Sigma}_k^{t+1^{-1}} = \boldsymbol{\Sigma}_k^{t^{-1}} + \frac{1}{N} \sum_{i=k}^{K} \beta_i \sum_{j=1}^{N} \left( \boldsymbol{\Sigma}_i^{t^{-1}} (\epsilon_i^j - \boldsymbol{\mu}_i^t)(\epsilon_i^j - \boldsymbol{\mu}_i^t)^\top \boldsymbol{\Sigma}_i^{t^{-1}} - \boldsymbol{\Sigma}_i^{t^{-1}} \right) (f_i(\bar{\epsilon}_i^j) - f_i(\bar{\boldsymbol{\mu}}_i^t)) \tag{15}$$

---

**Algorithm 1** BDTG

---

**Input:** Number of Batch Size $N$, step-size $\beta$, a pre-trained diffusion model $\widehat{\boldsymbol{\mu}}_\phi(\boldsymbol{x}_k, k)$, number of sampling step $K$, SDE solver coefficient $\tilde{\sigma}_k$ for $k \in \{1, \cdots, K\}$. Number of total iteration $T$.

**Initialization:** Initialize $\boldsymbol{\mu}_k^1 = \boldsymbol{0}, \boldsymbol{\Sigma}_k^1 = \boldsymbol{I}$, and set $\beta_k = \beta\tilde{\sigma}_k$ for $k \in \{1, \cdots, K\}$.

**for** $t = 1, \cdots, T$ **do**

    Take i.i.d. samples $\boldsymbol{x}_0^1, \cdots, \boldsymbol{x}_0^N \sim \mathcal{N}(\boldsymbol{0}, \boldsymbol{I})$.

    **for** $k = 1, \cdots, K$ **do**

        Take i.i.d. samples $\boldsymbol{\epsilon}_k^1, \cdots, \boldsymbol{\epsilon}_k^N \sim \mathcal{N}(\boldsymbol{\mu}_k^t, \boldsymbol{\Sigma}_k^t)$

        Set $\boldsymbol{x}_k^j = \widehat{\boldsymbol{\mu}}_\phi(\boldsymbol{x}_{k-1}^j, k-1) + \tilde{\sigma}_k \boldsymbol{\epsilon}_k^j$ for all $j \in \{1, \cdots, N\}$.

        Query black-box target function score $f_k(\bar{\boldsymbol{\epsilon}}_k^1) = F_k(\boldsymbol{x}_k^1), \cdots, f_k(\bar{\boldsymbol{\epsilon}}_k^N) = F_k(\boldsymbol{x}_k^N)$.

    **end for**

    Update $\boldsymbol{\mu}_k^{t+1}$ for all $k \in \{1, \cdots, K\}$ using Eq. (16)

    Update $\boldsymbol{\Sigma}_k^{t+1}$ for all $k \in \{1, \cdots, K\}$ using Eq. (17)

**end for**

---

where the offset term $f_i(\boldsymbol{\mu}_i^t)$ is employed to reduce variance.

In practice, to avoid the numeric scale problem, we normalize the score by $h(f_i(\bar{\boldsymbol{\epsilon}}_i^j)) = \frac{f_i(\bar{\boldsymbol{\epsilon}}_i^j) - \widehat{\mu}_i}{\widehat{\sigma}_i}$, where $\widehat{\mu}_i$ and $\widehat{\sigma}_i$ denote mean and standard deviation of function values $[f_i(\bar{\boldsymbol{\epsilon}}_i^1), \cdots, f_i(\bar{\boldsymbol{\epsilon}}_i^N)]$. We thus obtain the following update rule for practical updates.

$$\boldsymbol{\mu}_k^{t+1} = \boldsymbol{\mu}_k^t - \frac{1}{N} \sum_{i=k}^K \beta_i \sum_{j=1}^N [(\boldsymbol{\epsilon}_i^j - \boldsymbol{\mu}_i^t) \frac{f_i(\bar{\boldsymbol{\epsilon}}_i^j) - \widehat{\mu}_i}{\widehat{\sigma}_i}] \tag{16}$$

$$\boldsymbol{\Sigma}_k^{t+1^{-1}} = \boldsymbol{\Sigma}_k^{t^{-1}} + \frac{1}{N} \sum_{i=k}^K \beta_i \sum_{j=1}^N \left( \boldsymbol{\Sigma}_i^{t^{-1}} (\boldsymbol{\epsilon}_i^j - \boldsymbol{\mu}_i^t)(\boldsymbol{\epsilon}_i^j - \boldsymbol{\mu}_i^t)^\top \boldsymbol{\Sigma}_i^{t^{-1}} \right) \frac{f_i(\bar{\boldsymbol{\epsilon}}_i^j) - \widehat{\mu}_i}{\widehat{\sigma}_i} \tag{17}$$

Note that $\boldsymbol{\epsilon}_i^j = \boldsymbol{\mu}_i^t + \boldsymbol{\Sigma}_i^{t\frac{1}{2}} \boldsymbol{z}_i^j$ for $\boldsymbol{z}_i^j \sim \mathcal{N}(\boldsymbol{0}, \boldsymbol{I})$, Eq. (17) can be rewritten as

$$\boldsymbol{\Sigma}_k^{t+1^{-1}} = \boldsymbol{\Sigma}_k^{t^{-\frac{1}{2}}} (\boldsymbol{I} + \beta \boldsymbol{H}_k^t) \boldsymbol{\Sigma}_k^{t^{-\frac{1}{2}}} \tag{18}$$

where $\boldsymbol{H}_k^t$ is constructed as Eq. (19).

$$\boldsymbol{H}_k^t = \frac{1}{N} \sum_{i=k}^K \frac{\beta_i}{\beta} \sum_{j=1}^N \boldsymbol{z}_i^j \boldsymbol{z}_i^{j\top} \frac{f_i(\bar{\boldsymbol{\epsilon}}_i^j) - \widehat{\mu}_i}{\widehat{\sigma}_i} \tag{19}$$

The property of $\boldsymbol{H}_k^t$ has a crucial impact on the convergence speed.

We summarize our algorithm for black-box diffusion target generation algorithm (BDTG) into Algorithm 1. In Algorithm 1, the user preference can be incorporated via the black-box target score. In addition, $\tilde{\sigma}_k$ is the SDE solver coefficient. For example, when DPM solver (Lu et al., 2022a) is employed, $\tilde{\sigma}_k = \sigma_k \sqrt{e^{2h} - 1}$. More details about different solvers can be found in (Lu et al., 2022b).

## 5 CONVERGENCE ANALYSIS

In this section, we provide the convergence analysis of our general algorithm framework. Without loss of generality, we focus on minimizing the following sequential optimization problem:

$$\hat{F}(\bar{\boldsymbol{x}}_K) = \sum_{k=1}^K f_k(\bar{\boldsymbol{x}}_k) \tag{20}$$

**Remark:** The black-box function $f_k(\bar{\boldsymbol{x}}_k)$ with $\bar{\boldsymbol{x}}_k = [\boldsymbol{x}_1^\top, \cdots, \boldsymbol{x}_k^\top]^\top$ explicitly shows the dependency of the whole history trajectory $[\boldsymbol{x}_1, \cdots, \boldsymbol{x}_k]$. The sequential black-box optimization in Eq.(20) is general enough to include many interesting scenarios as special cases. One particular interesting problem is $f_k(\bar{\boldsymbol{x}}_k) = F(\boldsymbol{y}_k)$ where $\boldsymbol{y}_k$ is obtained by an unknown transition dynamic $\boldsymbol{y}_k = Q(\boldsymbol{y}_{k-1}, \boldsymbol{x}_k)$.

---

**Algorithm 2** SABSO Framework

---

**Input:** Batch-size $N$. Parameter $\beta_t, \alpha_t, \gamma_t$, and $\omega_t$. Number of total iteration $T$ and the step $K$.
**Initialization:** Initialize $\boldsymbol{\mu}_k^1 = \mathbf{0}, \boldsymbol{\Sigma}_k^1 = \tau \boldsymbol{I}$ for $k \in \{1, \cdots, K\}$.
**for** $t = 1, \cdots, T$ **do**
    **for** $k = 1, \cdots, K$ **do**
        Take i.i.d. samples $\boldsymbol{z}_k^1, \cdots, \boldsymbol{z}_k^N \sim \mathcal{N}(\mathbf{0}, \boldsymbol{I})$
        Set $\boldsymbol{x}_k^j = \boldsymbol{\mu}_k^t + \boldsymbol{\Sigma}_k^{t\frac{1}{2}} \boldsymbol{z}_k^j$ for $j \in \{1, \cdots, N\}$
        Query black-box objective function value $f_k(\bar{\boldsymbol{x}}_k^1), \cdots, f_k(\bar{\boldsymbol{x}}_k^N)$.
        Construct unbiased estimator $\hat{g}_{k1}, \cdots, \hat{g}_{kk}$ as Eq. (22).
        Compute $\hat{\mu}_k, \hat{\sigma}_k$ as the mean and std of $\{f_k(\bar{\boldsymbol{x}}_k^1), \cdots, f_k(\bar{\boldsymbol{x}}_k^N)\}$.
    **end for**
    **for** $k = 1, \cdots, K$ **do**
        Construct $\boldsymbol{H}_k^t = c_1 \frac{1}{N} \sum_{i=k}^K \sum_{j=1}^N \boldsymbol{z}_i^j {\boldsymbol{z}_i^j}^\top \frac{f_i(\bar{\boldsymbol{x}}_i^j) - \hat{\mu}_i}{\hat{\sigma}_i} + c_2 \boldsymbol{I}$ with constants $c_1 > 0, c_2 > 0$
        such that $\boldsymbol{H}_k^t \preceq \frac{1}{\alpha_t}(\frac{\beta_{t+1}}{\beta_t} - \omega_t)\boldsymbol{I} + \frac{\beta_{t+1}\gamma_t}{\alpha_t}\boldsymbol{\Sigma}_k^t$ and $\nu\boldsymbol{I} \preceq \hat{G}_k^t = \boldsymbol{\Sigma}_k^{t-\frac{1}{2}}\boldsymbol{H}_k^t\boldsymbol{\Sigma}_k^{t-\frac{1}{2}}$
        Set $\hat{G}_k^t = \boldsymbol{\Sigma}_k^{t-\frac{1}{2}}\boldsymbol{H}_k^t\boldsymbol{\Sigma}_k^{t-\frac{1}{2}}$.
        Set $\boldsymbol{\mu}_k^{t+1} = \boldsymbol{\mu}_k^t - \beta_t\boldsymbol{\Sigma}_k^t\big(\gamma_t\boldsymbol{\mu}_k^t + (\sum_{i=k}^K \hat{g}_{ik})\big)$
        Set $\boldsymbol{\Sigma}_k^{t+1^{-1}} = \omega_t\boldsymbol{\Sigma}_k^{t-1} + \alpha_t\hat{G}_k^t$.
    **end for**
**end for**

---

Instead of directly optimizing problem (20), we optimize an auxiliary problem (21) as

$$J(\bar{\boldsymbol{\mu}}_K, \bar{\boldsymbol{\Sigma}}_K) = \sum_{i=1}^K J_i(\bar{\boldsymbol{\mu}}_i, \bar{\boldsymbol{\Sigma}}_i) = \sum_{i=1}^K \mathbb{E}_{\bar{\boldsymbol{x}}_i \sim \mathcal{N}(\bar{\boldsymbol{\mu}}_i, \bar{\boldsymbol{\Sigma}}_i)}[f_i(\bar{\boldsymbol{x}}_i)] \tag{21}$$

where $J_i(\bar{\boldsymbol{\mu}}_i, \bar{\boldsymbol{\Sigma}}_i) = \mathbb{E}_{\bar{\boldsymbol{x}}_i \sim \mathcal{N}(\bar{\boldsymbol{\mu}}_i, \bar{\boldsymbol{\Sigma}}_i)}[f_i(\bar{\boldsymbol{x}}_i)]$.

Denote gradient estimator $\hat{g}_{ik}^t$ for the $i^{th}$ objective w.r.t. the $k^{th}$ component $\boldsymbol{\mu}_k$ at $t^{th}$ iteration as

$$\hat{g}_{ik}^t = \frac{1}{N}\sum_{j=1}^N \hat{g}_{ik}^{tj} = \frac{1}{N}\sum_{j=1}^N \boldsymbol{\Sigma}_k^{t-\frac{1}{2}}\boldsymbol{z}_k^j\big(f_i(\bar{\boldsymbol{\mu}}_i^t + \bar{\boldsymbol{\Sigma}}_i^{t\frac{1}{2}}\bar{\boldsymbol{z}}_i^j) - f_i(\bar{\boldsymbol{\mu}}_i^t)\big), \tag{22}$$

where $\hat{g}_{ik}^{tj}$ is the gradient estimator using $j^{th}$ i.i.d. sample:

$$\hat{g}_{ik}^{tj} = \boldsymbol{\Sigma}_k^{t-\frac{1}{2}}\boldsymbol{z}_k^j\big(f_i(\bar{\boldsymbol{\mu}}_i^t + \bar{\boldsymbol{\Sigma}}_i^{t\frac{1}{2}}\bar{\boldsymbol{z}}_i^j) - f_i(\bar{\boldsymbol{\mu}}_i^t)\big), \tag{23}$$

where $\boldsymbol{z}_1^j, \cdots, \boldsymbol{z}_i^j \sim \mathcal{N}(\mathbf{0}, \boldsymbol{I}_d)$ and $\bar{\boldsymbol{z}}_i^j = [\boldsymbol{z}_1^\top, \cdots, \boldsymbol{z}_i^\top]^\top$ for $i \geq k$.

We show our Stochastic Adaptive Black-box Sequential Optimization algorithm (SABSO) in Algorithm 2. Our SABSO can perform full matrix updates to take advantage of second-order information. We add a $\gamma_t\boldsymbol{\mu}_k^t$ term in the update step of $\boldsymbol{\mu}_k^{t+1}$ in Algorithm 2. The sequence $\gamma_t = O(\frac{1}{\sqrt{t+1}})$ decreasing to zero.

We now list the assumptions employed in our convergence analysis. All the assumptions are common in the literature. The assumptions are weak because neither smooth assumptions nor strongly convex assumptions are involved. Thus, our algorithm can handle non-smooth cases. More importantly, we do not add any additional assumptions of the auxiliary problem (21). This is important for practical use because we can not check whether the auxiliary problem satisfies the assumptions given a black-box original problem. To the best of our knowledge, our algorithm is the first full matrix adaptive black-box optimization algorithm that achieves a provable $O(\frac{d^2 K^4}{\sqrt{T}})$ convergence for convex functions without smooth and strongly convex assumptions, and any assumptions of the auxiliary problems.

**Assumption 5.1.** $f_1(\bar{\boldsymbol{x}}_1), \cdots, f_K(\bar{\boldsymbol{x}}_K)$ are all convex functions.

**Assumption 5.2.** $f_i(\bar{\boldsymbol{x}}_i)$ is a $L_i$-Lipschitz continuous function for $\forall i \in \{1, \cdots, K\}$, i.e., $|f_i(\bar{\boldsymbol{x}}_i) - f_i(\bar{\boldsymbol{y}}_i)| \leq L_i\|\bar{\boldsymbol{x}}_i - \bar{\boldsymbol{y}}_i\|_2$.

**Assumption 5.3.** The initialization $\bar{\theta}_K^1 := \{\bar{\mu}_K^1, \bar{\Sigma}_K^1\}$ is bounded, i.e., $\sum_{k=1}^{K} \|\mu_k^1 - \mu_k^*\|_{\Sigma_k^{1-1}}^2 \leq R$ and $\bar{\Sigma}_K^1 \in \mathcal{S}^+$, and $\underline{\tau} I \preceq \bar{\Sigma}_K^1 \preceq \bar{\tau} I$ for $\bar{\tau} \geq \underline{\tau} > 0$, and $\sum_{k=1}^{K} \|\mu_k^*\|_2^2 \leq B$.

**Theorem 5.4.** *Suppose the assumptions 5.1 5.2 5.3 holds. Set $\beta_t = t\beta$ with $\beta > 0$, $\alpha_t = \sqrt{t+1}\alpha$ with $\alpha > 0$, and $\gamma_t = \frac{\alpha\nu}{\beta\sqrt{t+1}}$, and $\nu > 0$, and $\omega_t = 1$. Initialize $\Sigma_k^1$ such that $\|\Sigma_k^1\|_2^{-1} \geq \frac{5}{3}\alpha\nu$ for $\forall k \in \{1, \cdots, K\}$. Then, running Algorithm 2 with $T$-steps, we have*

$$\frac{1}{T}\sum_{t=1}^{T}\sum_{k=1}^{K} f_k(\bar{\mu}_k^t) - \sum_{k=1}^{K} f_k(\bar{\mu}_k^*) \leq \frac{\sum_{k=1}^{K} \|\mu_k^1 - \mu_k^*\|_{\Sigma_k^{1-1}}^2}{2\beta T} + \frac{2\sqrt{T+1}C_1}{T} + \frac{4(T+1)^{\frac{1}{4}}C_2}{T} + \frac{\sqrt{T+2}C_3}{T} \tag{24}$$

$$\leq O(\frac{d^2 K^4}{\sqrt{T}}) \tag{25}$$

*where $\bar{\mu}_k^t = [\mu_1^{t\top}, \cdots, \mu_k^{t\top}]^\top$ and $\bar{\mu}_k^* = [\mu_1^{*\top}, \cdots, \mu_k^{*\top}]^\top$. And $C_1 = \frac{3\beta \sum_{i=1}^{K} KL_i^2(id+1)^2}{2\alpha\nu}$ and $C_2 = \frac{\sum_{i=1}^{K}\sqrt{3}idL_i}{\sqrt{\alpha\nu}}$, $C_3 = \frac{\alpha\nu B}{\beta}$.*

Detailed proof can be found in Appendix D. In Theorem 5.4, the error term $\frac{\sqrt{T+2}C_3}{T}$ in Eq.(24) is due to the $\gamma_t$-enlargement. Error term $\frac{4(T+1)^{\frac{1}{4}}C_2}{T}$ results from the covariance matrix term $\bar{\Sigma}_K$ in Gaussian-smooth relaxation of the original problem. The first two error terms result from the stochastic gradient update.

Note that for a convex function $f(x)$, we have $f(\frac{1}{T}\sum_{t=1}^{T} x_t) \leq \frac{1}{T}\sum_{t=1}^{T} f(x_t)$. Then, we can directly obtain the solution of the original problem (20) by averaging the auxiliary variable $\mu_K^t$ for the auxiliary problem (21).

**Corollary 5.5.** *Suppose the assumptions 5.1 5.2 5.3 holds. Set $\beta_t = t\beta$ with $\beta > 0$, $\alpha_t = \sqrt{t+1}\alpha$ with $\alpha > 0$, and $\gamma_t = \frac{\alpha\nu}{\beta\sqrt{t+1}}$, and $\nu > 0$, and $\omega_t = 1$. Initialize $\Sigma_k^1$ such that $\|\Sigma_k^1\|_2^{-1} \geq \frac{5}{3}\alpha\nu$ for $\forall k \in \{1, \cdots, K\}$. Then, running Algorithm 2 with $T$-steps, set $\bar{x}_K^T = \frac{1}{T}\sum_{t=1}^{T} \bar{\mu}_K^t$, we have the cumulative regret as:*

$$\sum_{k=1}^{K} f_k(\bar{x}_k^T) - \sum_{k=1}^{K} f_k(\bar{\mu}_k^*) \leq \frac{\sum_{k=1}^{K} \|\mu_k^1 - \mu_k^*\|_{\Sigma_k^{1-1}}^2}{2\beta T} + \frac{2\sqrt{T+1}C_1}{T} + \frac{4(T+1)^{\frac{1}{4}}C_2}{T} + \frac{\sqrt{T+2}C_3}{T} \tag{26}$$

$$\leq O(\frac{d^2 K^4}{\sqrt{T}}) \tag{27}$$

*where $\bar{x}_k^T = [x_1^{T\top}, \cdots, x_k^{T\top}]^\top$ and $\bar{\mu}_k^* = [\mu_1^{*\top}, \cdots, \mu_k^{*\top}]^\top$. And $C_1 = \frac{3\beta \sum_{i=1}^{K} KL_i^2(id+1)^2}{2\alpha\nu}$ and $C_2 = \frac{\sum_{i=1}^{K}\sqrt{3}idL_i}{\sqrt{\alpha\nu}}$, $C_3 = \frac{\alpha\nu B}{\beta}$.*

**Remark:** Note that for convex problems, the optimum $\{\bar{\mu}_K^*, 0\}$ of the auxiliary problem (21) is also the optimum of the original problem (20), i.e., $J(\bar{\mu}_K^*, 0) = \hat{F}(\bar{\mu}_K^*)$. Thus, the solution $\bar{x}_k^T$ achieve a $O(\frac{d^2 K^4}{\sqrt{T}})$ cumulative regret of the original problem (20). In addition, our algorithm can handle non-smooth problems without expert designing of proximal operators for different types of non-smooth functions. This can be remarkably interesting when the unknown function involves compositions of lots of different types of non-smooth functions, in which case human experts can not derive the operators explicitly.

## 6 EXPERIMENTS

### 6.1 EMPIRICAL STUDY ON NUMERICAL TEST PROBLEM

We first evaluate our algorithm on minimizing the numerical cumulative summation problem $\hat{F}(\bar{x}_K) = \sum_{i=1}^{d} f(x_k)$ with black-box transition dynamic $x_k = x_{k-1} + 0.1$. We test two cases: $L_2$-norm Ellipsoid $f(x) := \sum_{m=1}^{d} 10^{\frac{6(m-1)}{d-1}} x_m^2$ and $L_1$-norm Ellipsoid $f(x) := \sum_{m=1}^{d} 10^{\frac{6(m-1)}{d-1}} |x_m|$

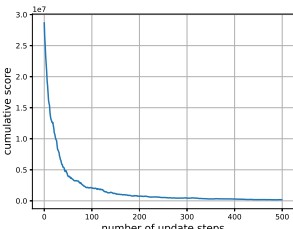 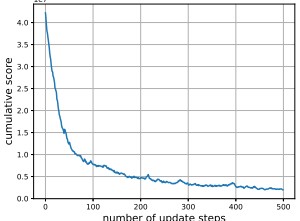 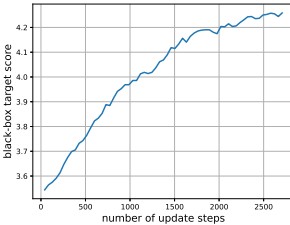

(a) Cumulative $L_2$-norm Ellipsoid problem  (b) Cumulative $L_1$-norm Ellipsoid problem  (c) Diffusion model Black-box Fine-tuning

Figure 1: Target score v.s. the number of update steps on different problems

Table 1: Black-box function scores for each method evaluated on 2700 generated images

| | Mean | Std | 50%ile | 80%ile | 95%ile |
|---|---|---|---|---|---|
| Dataset | 0.0000 | 1.0000 | 0.0511 | 0.7876 | 1.7414 |
| Conditional model | 3.5048 | 0.6672 | 3.5065 | 4.0601 | 4.5863 |
| DDOM | 3.6571 | 0.6252 | 3.6764 | 4.1807 | 4.6631 |
| Fine-tune (steps=1596) (ours) | 4.0998 | 0.5976 | 4.1259 | 4.6083 | 5.0468 |
| Fine-tune (steps=2646) (ours) | 4.2287 | 0.5642 | 4.2739 | 4.7069 | 5.0797 |
| Extend dataset (ours) | 4.7948 | 0.4897 | 4.8166 | 5.2222 | 5.5511 |

The plot of the target score is shown in Figure 1 (a) and (b), respectively. We can see our converge fast in the sequential optimization test problem.

## 6.2 EMPIRICAL STUDY ON BLACK-BOX DIFFUSION TARGET GENERATION

**Black-box Target Score:** We employ the CLIP model[1] (Radford et al., 2021) to compute the black-box target score. Specifically, we compute the cosine distance between the latent embedding of the generated image and the latent embedding of the target text and employ the normalized cosine distance $\frac{y-\mu_y}{\sigma_y}$ as the black-box target score, where $\mu_y$ and $\sigma_y$ denotes the mean and standard deviation of the cosine distances between the target and the images in the dataset. We chose "a close-up of a man with long hair" as our target text to compute the black-box target score. The larger the score is the better.

**Baselines:** We compare our methods with the DDOM generation (Krishnamoorthy et al., 2023), the conditional model generation conditioned on the pre-computed max score from the dataset, and the trivial max score from the dataset as baselines. In addition to our black-box methods optimization for diffusion fine-tuning, we employ the generated images from our fine-tuned diffusion model to extend the dataset and employ the extended dataset to retrain a conditional model.

**Dataset and Implementation:** In our experiments, we use the dataset CelebA-HQ[2] that contains 30,000 face images. We employ DPM-Solver++ (Lu et al., 2022a) as the sampler for all experiments in both the training and evaluation phases. In all experiments, we set the number of sampling steps as $K = 14$. More implementation details can be found in the Appendix E.

For each method, we generate 2700 images and evaluate the normalized scores. The mean, std, 50%-Percentile score, 80%-Percentile score and 95%-Percentile score of each method are shown in Table 1. We can observe that both our fine-tuning method and extended dataset retraining method achieve significantly higher scores compared with baselines consistently across all metrics. Moreover, we can see that re-training with our extended dataset achieves the highest scores among all the compared methods, which demonstrates the ability and potential of our method for target generation. In addition, all the methods obtain significantly higher scores compared with the training dataset.

We further show the plot of the relationship between the black-box function score value and the number of update steps of our method in Figure 1 (c). We observe that the score increases almost linearly with the number of update steps, which demonstrates the potential of our optimization method in diffusion target generation.

---

[1]https://huggingface.co/sentence-transformers/clip-ViT-L-14
[2]https://huggingface.co/datasets/huggan/CelebA-HQ

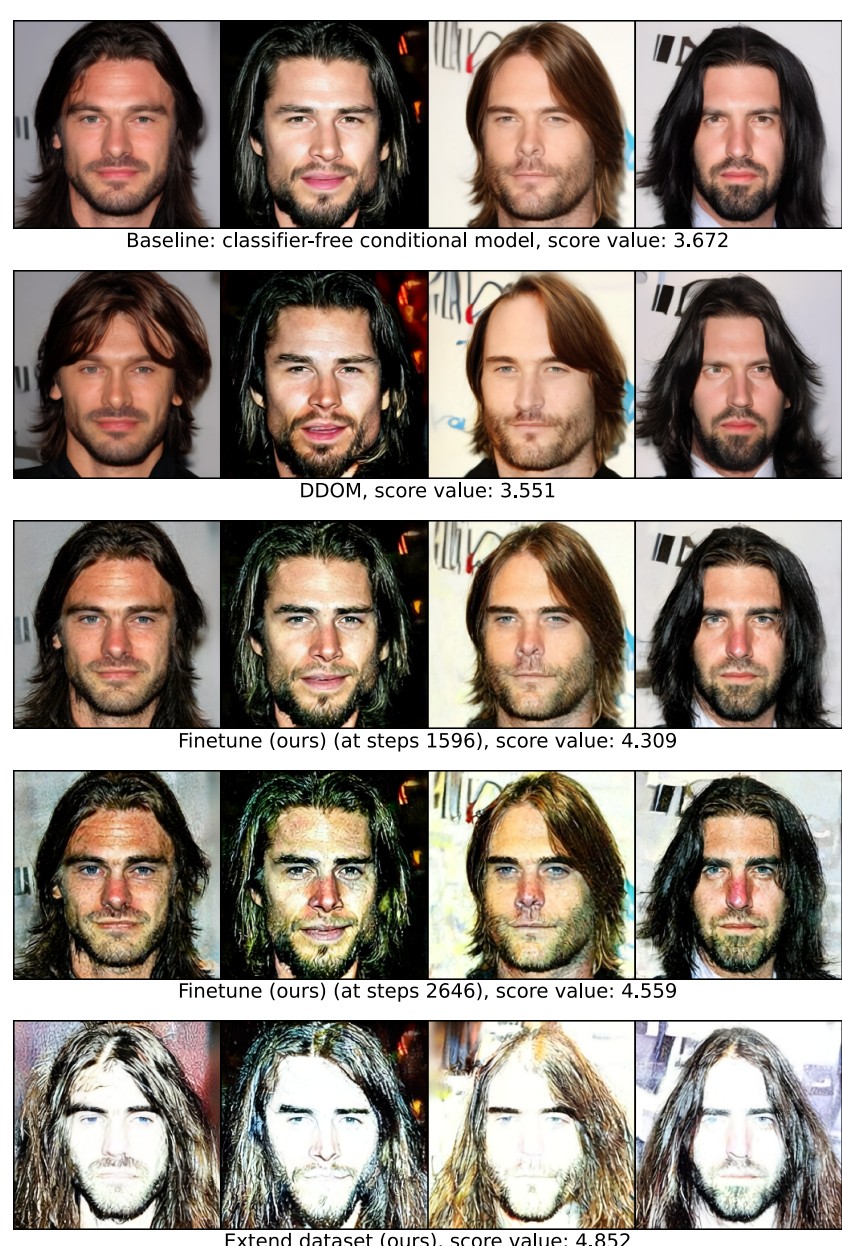

Baseline: classifier-free conditional model, score value: 3.672

DDOM, score value: 3.551

Finetune (ours) (at steps 1596), score value: 4.309

Finetune (ours) (at steps 2646), score value: 4.559

Extend dataset (ours), score value: 4.852

Figure 2: Generated Images of all methods

We further visualize the images generated by different methods with the same initialized noise $x_0$ in Figure 2. We observe that our method trades off optimizing the score at the cost of image quality. This issue may be mitigated by incorporating quality measurement into the black-box function. We leave it to future work.

## 7 CONCLUSION

In this paper, we formulated the fine-tuning of the diffusion model for black-box target generation as a sequential black-box optimization problem. We proposed a novel stochastic adaptive sequential black-box optimization(SASBO) algorithm to address this problem. Theoretically, we prove a $O(\frac{d^2 K^4}{\sqrt{T}})$ convergence rate of SASBO without smooth and strongly convex assumptions. Thus, our theoretical results hold true for all non-smooth/smooth convex function families that are of great challenge for full matrix adaptive black-box algorithms to converge. Empirically, our method enables the fine-tuning of the diffusion model to generate targeted images with a high black-box target score.

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

# APPENDIX

## A  ADDITIONAL EXPERIMENTS

### A.1  TARGETED IMAGE GENERATION

We evaluate our revised method on two targeted image generation cases: (1) "long hair man" (2)"Asian face". The "Asian face" is rare in the dataset. As a result, the targeted generation focuses more on out-of-distribution generation, which is more challenging than the "long hair man" case. For the "long hair man" case, we keep the target text as "a close-up of a man with long hair", which is the same as in our previous submission version. For the "Asian face" case, we set the target text as "a high quality close up of an asian".

For both cases, we set the number of iterations $T$ of our method as $T = 120$. The experimental results reported are at $T = 120$. We perform independent draws to generate $3,000$ images for evaluation. The same set consists of $3,000$ i.i.d. sampled initial noise $x_0 \sim \mathcal{N}(0, I)$ is employed for all the methods to generate images for comparison.

#### A.1.1  TARGETED IMAGE GENERATION: LONG HAIR MAN

In this experiment, the target text is set as "a close-up of a man with long hair". Figure 3 shows the comparison of generated images. Figure 4 shows 64 randomly generated images by our method. Table 2 shows the score values.

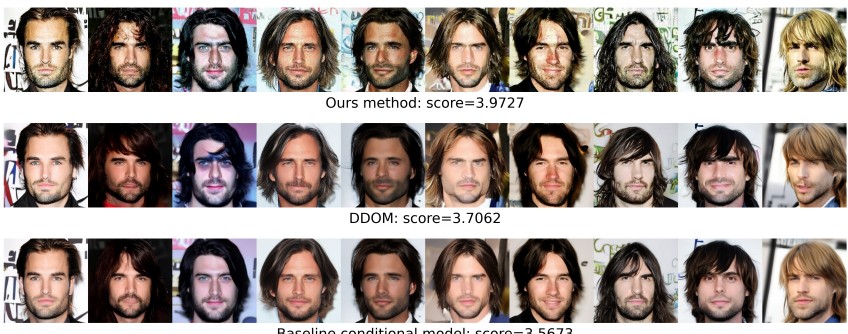

Ours method: score=3.9727

DDOM: score=3.7062

Baseline conditional model: score=3.5673

Figure 3: Comparison of different methods on "long hair man"

Table 2: Score values evaluated on 3000 generated images on "long hair man"

|  | Mean | Std | 50%ile | 80%ile | 95%ile |
|---|---|---|---|---|---|
| Dataset | 0.0000 | 1.0000 | 0.0511 | 0.7876 | 1.7414 |
| Conditional model | 3.5673 | 0.6762 | 3.5672 | 4.1299 | 4.6761 |
| DDOM | 3.7062 | 0.6372 | 3.7110 | 4.2618 | 4.7319 |
| Our method | **3.9727** | **0.5902** | **3.9937** | **4.4598** | **4.9197** |

#### A.1.2  TARGETED IMAGE GENERATION: ASIAN FACE

In this experiment, the target text is set as "a high quality close up of an asian". Figure 5 shows the comparison of generated images. Figure 6 shows 64 randomly generated images by our method. Table 3 shows the score values.

Table 3: Score values evaluated on 3000 generated images on "Asian Face"

|  | Mean | Std | 50%ile | 80%ile | 95%ile |
|---|---|---|---|---|---|
| Dataset | 0.0000 | 1.0000 | 0.0511 | 0.7876 | 1.7414 |
| Conditional model | 2.8784 | 0.5654 | 2.8870 | 3.3682 | 3.7874 |
| DDOM | 3.2397 | 0.4824 | 3.2693 | 3.6346 | 4.0078 |
| Our method | **3.2435** | **0.5453** | **3.2733** | **3.7074** | **4.0789** |

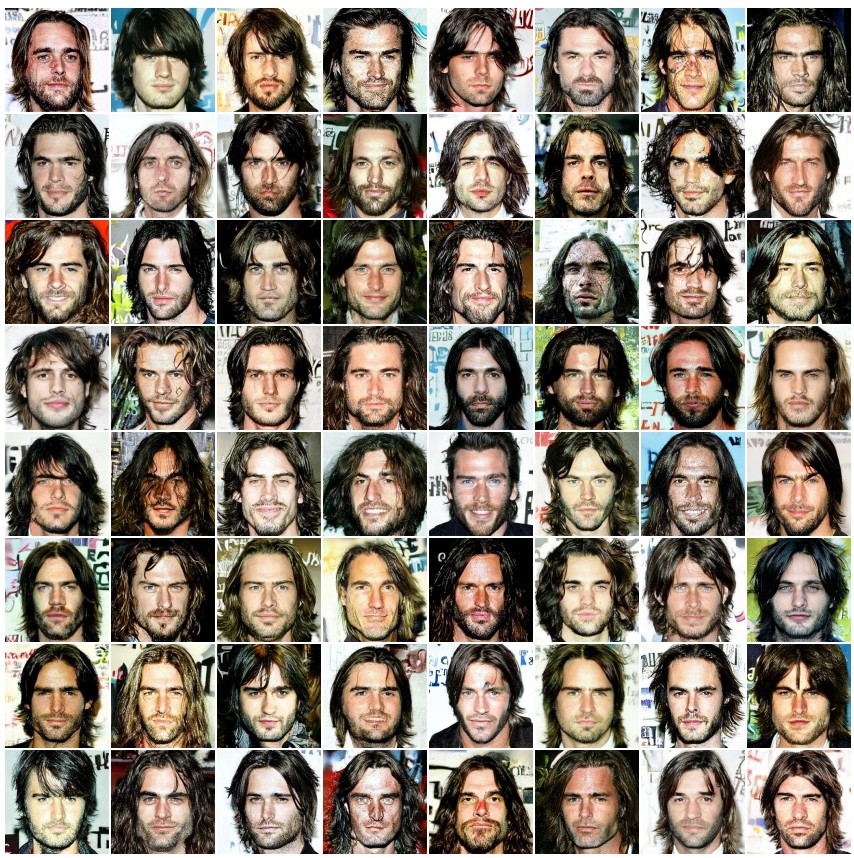

Figure 4: 64 images generated by our method for "long hair man"

Table 4: Normalized maximum score on Design Bench evaluated on 5 independent runs

|                   | SUPERCON.             | CHEMBL                |
|-------------------|-----------------------|-----------------------|
| Conditional model | $0.4824 \pm 0.0466$   | $0.6345 \pm 0.0026$   |
| DDOM              | $0.4777 \pm 0.0350$   | $0.6344 \pm 0.0027$   |
| Our method        | $\mathbf{0.5123 \pm 0.0145}$ | $\mathbf{0.6392 \pm 0.0055}$ |

We can see that the visual quality of our method's generated images is better than DDOM's on the "Asian face" cases. DDOM employs a reweighed sampling of training samples to train a conditional diffusion model from scratch. This training scheme focuses on the tail of the distribution, which is more vulnerable to overfitting, especially for out-of-distribution target generation cases where the target image is rare in the training set.

## A.2 DESIGN-BENCH: SUPERCONDUCTOR

We compare our method with the DDOM's on the design-bench tasks. We train our model for $T = 200$ iterations. We follow Krishnamoorthy et al. (2023) to perform 5 independent runs with different seed, and report the normalized maximum score along with standard deviation. The normalization method is the same as Krishnamoorthy et al. (2023). We experiment on two tasks, Superconductor and ChEMBL. Experimental results are shown in Table 4. Our method outperforms the DDOM on both tasks, which shows the potential of our method on different domains beyond the targeted image generation.

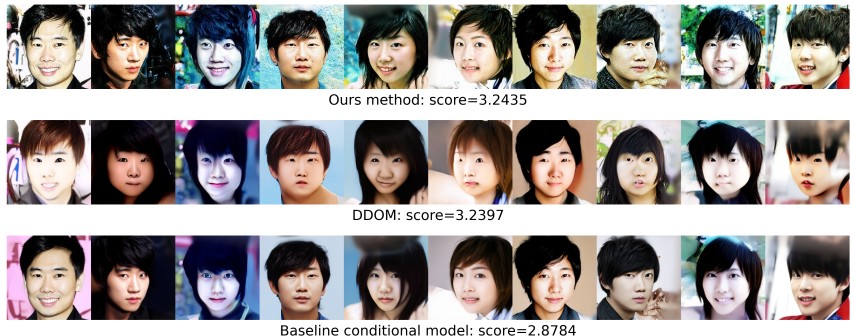

Figure 5: Comparison of different methods on "Asian Face"

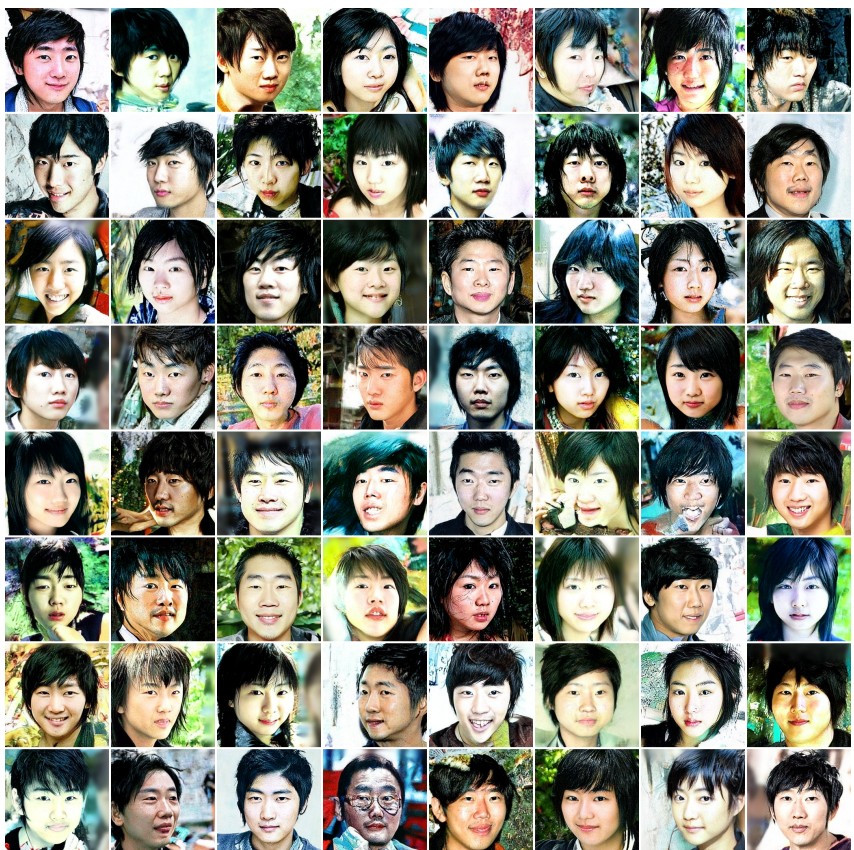

Figure 6: 64 images generated by our method for "Asian Face"

# B DERIVATION OF UPDATE RULE

The minimization can be rewritten as

$$
\sum_{k=1}^{K} \left\langle \bar{\theta}_k - \bar{\theta}_k^t, \beta_k \nabla_{\bar{\theta}_k^t} J_k(\bar{\theta}_k^t) \right\rangle + \mathrm{KL}(q_\theta \| q_{\theta^t}) = \mathbb{E}_{\boldsymbol{x}_0 \sim \mathcal{N}(\mathbf{0}, \boldsymbol{I})} \sum_{k=1}^{K} \beta_k \bar{\boldsymbol{\mu}}_k^{t\top} \nabla_{\bar{\boldsymbol{\mu}}_k^t} \mathbb{E}_{q_{\bar{\theta}_k^t}}[f_k(\bar{\boldsymbol{x}}_k)] +
$$

$$
\sum_{k=1}^{K} \beta_k \mathrm{tr}(\bar{\Sigma}_k \nabla_{\bar{\boldsymbol{\Sigma}}_k^t} \mathbb{E}_{q_{\bar{\theta}_k^t}}[f_k(\bar{\boldsymbol{x}}_k)]) + \frac{1}{2} \left[ \mathrm{tr}(\Sigma_K^{t\,-1} \bar{\Sigma}_K) + (\bar{\boldsymbol{\mu}}_K - \bar{\boldsymbol{\mu}}_K^t)^\top \Sigma_K^{t\,-1} (\bar{\boldsymbol{\mu}}_K - \bar{\boldsymbol{\mu}}_K^t) + \log \frac{|\bar{\Sigma}_K^t|}{|\bar{\Sigma}_K|} - d \right],
$$

$$
(28)
$$

where $\nabla_{\bar{\boldsymbol{\mu}}_k^t}\mathbb{E}_{q_{\bar{\theta}_k^t}}[f_k(\bar{\boldsymbol{x}}_k)]$ and $\nabla_{\bar{\boldsymbol{\Sigma}}_k^t}\mathbb{E}_{q_{\bar{\theta}_k^t}}[f_k(\bar{\boldsymbol{x}}_k)]$ denotes the derivative w.r.t $\bar{\boldsymbol{\mu}}_k$ and $\bar{\boldsymbol{\Sigma}}_k$ taking at $\bar{\boldsymbol{\mu}}_k = \bar{\boldsymbol{\mu}}_k^t$ and $\bar{\boldsymbol{\Sigma}}_k = \bar{\boldsymbol{\Sigma}}_k^k$. The above problem is convex with respect to $\bar{\boldsymbol{\mu}}_K = [\boldsymbol{\mu}_1^\top, \cdots, \boldsymbol{\mu}_K^\top]^\top$ and $\bar{\boldsymbol{\Sigma}}_K = diag(\boldsymbol{\Sigma}_1, \cdots, \boldsymbol{\Sigma}_K)$. Taking the derivative w.r.t $\bar{\boldsymbol{\mu}}_K$ and $\bar{\boldsymbol{\Sigma}}_K$ and setting them to zero, for $k^{th}$ component , we can obtain that

$$\mathbb{E}_{\boldsymbol{x}_0 \sim \mathcal{N}(\boldsymbol{0}, \boldsymbol{I})}\sum_{i=k}^K \beta_i \nabla_{\boldsymbol{\mu}_k^t}\mathbb{E}_{q_{\bar{\theta}_i^t}}[f_i(\bar{\boldsymbol{x}}_i)] + \boldsymbol{\Sigma}_k^{t\,-1}(\boldsymbol{\mu}_k - \boldsymbol{\mu}_k^t) = 0 \tag{29}$$

$$\mathbb{E}_{\boldsymbol{x}_0 \sim \mathcal{N}(\boldsymbol{0}, \boldsymbol{I})}\sum_{i=k}^K \beta_i \nabla_{\boldsymbol{\Sigma}_k^t}\mathbb{E}_{q_{\bar{\theta}_i^t}}[f_i(\bar{\boldsymbol{x}}_i)] + \frac{1}{2}[\boldsymbol{\Sigma}_k^{-1} - \boldsymbol{\Sigma}_k^{t\,-1}] = 0. \tag{30}$$

for $k \in \{1, \cdots, K\}$.

Set $\boldsymbol{\mu}_k^{t+1} = \boldsymbol{\mu}_k$ and $\boldsymbol{\Sigma}_k^{t+1\,-1} = \boldsymbol{\Sigma}_k^{-1}$ in the above equation. We then have the update rule as

$$\boldsymbol{\mu}_k^{t+1} = \boldsymbol{\mu}_k^t - \mathbb{E}_{\boldsymbol{x}_0 \sim \mathcal{N}(\boldsymbol{0}, \boldsymbol{I})}\sum_{i=k}^K \beta_i \boldsymbol{\Sigma}_k^t \nabla_{\boldsymbol{\mu}_k^t}\mathbb{E}_{q_{\bar{\theta}_i^t}}[f_i(\bar{\boldsymbol{x}}_i)] \tag{31}$$

$$\boldsymbol{\Sigma}_k^{t+1\,-1} = \boldsymbol{\Sigma}_k^{t\,-1} + \mathbb{E}_{\boldsymbol{x}_0 \sim \mathcal{N}(\boldsymbol{0}, \boldsymbol{I})}\sum_{i=k}^K 2\beta_i \nabla_{\boldsymbol{\Sigma}_k^t}\mathbb{E}_{q_{\bar{\theta}_i^t}}[f_i(\bar{\boldsymbol{x}}_i)]. \tag{32}$$

for $k \in \{1, \cdots, K\}$.

In addition, note that the gradient has the following closed-form (Wierstra et al., 2014)

$$\nabla_{\boldsymbol{\mu}_k^t}\mathbb{E}_{q_{\bar{\theta}_i^t}}[f_i(\bar{\boldsymbol{x}}_i)] = \boldsymbol{\Sigma}_k^{t\,-1}\mathbb{E}_{q_{\bar{\theta}_i^t}}[(\boldsymbol{x}_k - \boldsymbol{\mu}_k)f_i(\bar{\boldsymbol{x}}_i)] \tag{33}$$

$$\nabla_{\boldsymbol{\Sigma}_k^t}\mathbb{E}_{q_{\bar{\theta}_i^t}}[f_i(\bar{\boldsymbol{x}}_i)] = \frac{1}{2}\mathbb{E}_{q_{\bar{\theta}_i^t}}[\left(\boldsymbol{\Sigma}_k^{t\,-1}(\boldsymbol{x}_k - \boldsymbol{\mu}_k)(\boldsymbol{x}_k - \boldsymbol{\mu}_k)^\top \boldsymbol{\Sigma}_k^{t\,-1} - \boldsymbol{\Sigma}_k^{t\,-1}\right)(f_i(\bar{\boldsymbol{x}}_i))] \tag{34}$$

Then, we have that

$$\boldsymbol{\mu}_k^{t+1} = \boldsymbol{\mu}_k^t - \sum_{i=k}^K \beta_i \mathbb{E}_{\boldsymbol{x}_0 \sim \mathcal{N}(\boldsymbol{0}, \boldsymbol{I})}\mathbb{E}_{q_{\bar{\theta}_i^t}}[(\boldsymbol{x}_k - \boldsymbol{\mu}_k)f_i(\bar{\boldsymbol{x}}_i)] \tag{35}$$

$$\boldsymbol{\Sigma}_k^{t+1\,-1} = \boldsymbol{\Sigma}_k^{t\,-1} + \sum_{i=k}^K \beta_i \mathbb{E}_{\boldsymbol{x}_0 \sim \mathcal{N}(\boldsymbol{0}, \boldsymbol{I})}\mathbb{E}_{q_{\bar{\theta}_i^t}}[\left(\boldsymbol{\Sigma}_k^{t\,-1}(\boldsymbol{x}_k - \boldsymbol{\mu}_k)(\boldsymbol{x}_k - \boldsymbol{\mu}_k)^\top \boldsymbol{\Sigma}_k^{t\,-1} - \boldsymbol{\Sigma}_k^{t\,-1}\right)(f_i(\bar{\boldsymbol{x}}_i))]. \tag{36}$$

## C   TECHNICAL LEMMAS

In this section, we introduce the following technical lemmas for convergence analysis.

**Lemma C.1.** *Given a positive definite matrix $\boldsymbol{\Sigma}$, we have $\|\boldsymbol{\Sigma}(\boldsymbol{x}+\boldsymbol{y})\|_{\boldsymbol{\Sigma}^{-1}}^2 \leq 2(\|\boldsymbol{\Sigma}\boldsymbol{x}\|_{\boldsymbol{\Sigma}^{-1}}^2 + \|\boldsymbol{\Sigma}^{\frac{1}{2}}\boldsymbol{y}\|_2^2)$*

*Proof.*

$$\|\boldsymbol{\Sigma}(\boldsymbol{x} + \boldsymbol{y})\|_{\boldsymbol{\Sigma}^{-1}}^2 = \left\langle \boldsymbol{\Sigma}^{-1}\boldsymbol{\Sigma}(\boldsymbol{x} + \boldsymbol{y}), \boldsymbol{\Sigma}(\boldsymbol{x} + \boldsymbol{y})\right\rangle \tag{37}$$

$$= \left\langle (\boldsymbol{x} + \boldsymbol{y}), \boldsymbol{\Sigma}(\boldsymbol{x} + \boldsymbol{y})\right\rangle \tag{38}$$

$$= \|\boldsymbol{\Sigma}^{\frac{1}{2}}\boldsymbol{x} + \boldsymbol{\Sigma}^{\frac{1}{2}}\boldsymbol{y}\|_2^2 \tag{39}$$

$$\leq 2(\|\boldsymbol{\Sigma}^{\frac{1}{2}}\boldsymbol{x}\|_2^2 + \|\boldsymbol{\Sigma}^{\frac{1}{2}}\boldsymbol{y}\|_2^2) \tag{40}$$

Note that $\|\boldsymbol{\Sigma}^{\frac{1}{2}}\boldsymbol{x}\|_2^2 = \left\langle \boldsymbol{x}, \boldsymbol{\Sigma}\boldsymbol{x}\right\rangle = \left\langle \boldsymbol{\Sigma}^{-1}\boldsymbol{\Sigma}\boldsymbol{x}, \boldsymbol{\Sigma}\boldsymbol{x}\right\rangle = \|\boldsymbol{\Sigma}\boldsymbol{x}\|_{\boldsymbol{\Sigma}^{-1}}^2$, we achieve that

$$\|\boldsymbol{\Sigma}(\boldsymbol{x} + \boldsymbol{y})\|_{\boldsymbol{\Sigma}^{-1}}^2 \leq 2(\|\boldsymbol{\Sigma}\boldsymbol{x}\|_{\boldsymbol{\Sigma}^{-1}}^2 + \|\boldsymbol{\Sigma}^{\frac{1}{2}}\boldsymbol{y}\|_2^2) \tag{41}$$

$\square$

**Lemma C.2.** *Suppose the gradient estimator $\hat{g}_{ik}^t$ for the $i^{th}$ objective w.r.t. the $k^{th}$ component $\boldsymbol{\mu}_k$ at $t^{th}$ iteration as*

$$\hat{g}_{ik}^t = \boldsymbol{\Sigma}_k^{t\,-\frac{1}{2}} \boldsymbol{z}_k \big( f_i(\bar{\boldsymbol{\mu}}_i^t + \bar{\boldsymbol{\Sigma}}_i^{t\,\frac{1}{2}} \bar{\boldsymbol{z}}_i) - f_i(\bar{\boldsymbol{\mu}}_i^t) \big), \tag{42}$$

*where $\boldsymbol{z}_1, \cdots, \boldsymbol{z}_i \sim \mathcal{N}(\boldsymbol{0}, \boldsymbol{I}_d)$ and $\bar{\boldsymbol{z}}_i = [\boldsymbol{z}_1^\top, \cdots, \boldsymbol{z}_i^\top]^\top$, $i \geq k$. Suppose assumptions 5.2 hold, using the parameter setting in Theorem 5.4, and the covariance matrix update $\hat{G}_k^t = \boldsymbol{\Sigma}_k^{t\,-\frac{1}{2}} \boldsymbol{H}_k^t \boldsymbol{\Sigma}_k^{t\,-\frac{1}{2}}$ are positive semi-definite matrix that satisfies $\nu \boldsymbol{I} \preceq \hat{G}_k^t$. Apply the update rule $\boldsymbol{\Sigma}_k^{t+1\,-1} = \omega_t \boldsymbol{\Sigma}_k^{t\,-1} + \alpha_t \hat{G}_k^t$, we have*

*(a) $\hat{g}_{ik}^t$ is an unbiased estimator of $g_{ik}^t = \nabla_{\boldsymbol{\mu}_k} \mathbb{E}_{\bar{\boldsymbol{x}} \sim \mathcal{N}(\bar{\boldsymbol{\mu}}_i^t, \bar{\boldsymbol{\Sigma}}_i^t)}[f_i(\bar{\boldsymbol{x}})]$.*

*(b) $\|\boldsymbol{\Sigma}_k^{t+1}\|_2 \leq \frac{1}{\|\boldsymbol{\Sigma}_k^t\|_2^{-1} + \sqrt{t+1}\alpha\nu} \leq \cdots \leq \frac{3}{2\alpha\nu} \frac{1}{(t+1)^{\frac{3}{2}} + \frac{3}{2}} \leq \frac{3}{2\alpha\nu} \frac{1}{(t+1)^{\frac{3}{2}}}$.*

*(c) $\mathbb{E} \sum_{k=1}^K \|\boldsymbol{\Sigma}_k^t (\sum_{i=k}^K \hat{g}_{ik})\|_{\boldsymbol{\Sigma}_k^{t\,-1}}^2 \leq \frac{3 \sum_{i=1}^K KL_i^2 (id+1)^2}{2t^{\frac{3}{2}}\alpha\nu}$*

*For the average of i.i.d. sampled unbiased gradient estimators (each one has the same form as Eq.(42)), the results (a),(b),(c) still hold true.*

*Proof.* (a). We first show that $\hat{g}_{ik}^t$ is an unbiased estimator of $\nabla_{\boldsymbol{\mu}_k} \mathbb{E}_{\bar{\boldsymbol{x}} \sim \mathcal{N}(\bar{\boldsymbol{\mu}}_i^t, \bar{\boldsymbol{\Sigma}}_i^t)}[f_i(\bar{\boldsymbol{x}})]$.

$$\mathbb{E}_{\bar{\boldsymbol{z}}_i}[\hat{g}_{ik}^t] = \mathbb{E}_{\bar{\boldsymbol{z}}_i}[\boldsymbol{\Sigma}_k^{t\,-\frac{1}{2}} \boldsymbol{z}_k f_i(\bar{\boldsymbol{\mu}}_i^t + \bar{\boldsymbol{\Sigma}}_i^{t\,\frac{1}{2}} \bar{\boldsymbol{z}}_i)] - \mathbb{E}_{\bar{\boldsymbol{z}}_i}[\boldsymbol{\Sigma}_k^{t\,-\frac{1}{2}} \boldsymbol{z}_k f_i(\bar{\boldsymbol{\mu}}_i^t)] \tag{43}$$

$$= \mathbb{E}_{\bar{\boldsymbol{z}}_i}[\boldsymbol{\Sigma}_k^{t\,-\frac{1}{2}} \boldsymbol{z}_k f_i(\bar{\boldsymbol{\mu}}_i^t + \bar{\boldsymbol{\Sigma}}_i^{t\,\frac{1}{2}} \bar{\boldsymbol{z}}_i)] \tag{44}$$

$$= \mathbb{E}_{\bar{\boldsymbol{x}} \sim \mathcal{N}(\bar{\boldsymbol{\mu}}_i^t, \bar{\boldsymbol{\Sigma}}_i^t)}[\boldsymbol{\Sigma}_k^{t\,-1}(\boldsymbol{x}_k - \boldsymbol{\mu}_k^t) f_i(\bar{\boldsymbol{x}})] \tag{45}$$

Note that $\bar{\boldsymbol{\Sigma}}_i^t = diag(\boldsymbol{\Sigma}_1^t, \cdots, \boldsymbol{\Sigma}_i^t)$ is a block-wise diagonal matrix, and $\bar{\boldsymbol{\mu}}_i = [\boldsymbol{\mu}_1^\top, \cdots, \boldsymbol{\mu}_i^\top]^\top$, $i \geq k$ , we then have that

$$\mathbb{E}_{\bar{\boldsymbol{z}}_i}[\hat{g}_{ik}^t] = \mathbb{E}_{\bar{\boldsymbol{x}} \sim \mathcal{N}(\bar{\boldsymbol{\mu}}_i^t, \bar{\boldsymbol{\Sigma}}_i^t)}[\boldsymbol{\Sigma}_k^{t\,-1}(\boldsymbol{x}_k - \boldsymbol{\mu}_k^t) f_i(\bar{\boldsymbol{x}})] = \nabla_{\boldsymbol{\mu}_k} \mathbb{E}_{\bar{\boldsymbol{x}} \sim \mathcal{N}(\bar{\boldsymbol{\mu}}_i^t, \bar{\boldsymbol{\Sigma}}_i^t)}[f_i(\bar{\boldsymbol{x}})]. \tag{46}$$

For $N$ i.i.d. sampled unbiased gradient estimator, the average is still an unbiased gradient estimator. $\square$

*Proof.* (b) We now prove the decay of the spectral norm of covariance matrix.

From the update rule of $\boldsymbol{\Sigma}_k^t$, we know that

$$\boldsymbol{\Sigma}_k^{t+1\,-1} = \omega_t \boldsymbol{\Sigma}_k^{t\,-1} + \alpha_t \hat{G}_k^t \tag{47}$$

Note that $\nu \boldsymbol{I} \preceq \hat{G}_k^t$, we have that

$$\lambda_{\min}\big(\boldsymbol{\Sigma}_k^{t+1\,-1}\big) = \lambda_{\min}\big(\omega_t \boldsymbol{\Sigma}_k^{t\,-1} + \alpha_t \hat{G}_k^t\big) \tag{48}$$

$$\geq \omega_t \lambda_{\min}\big(\boldsymbol{\Sigma}_k^{t\,-1}\big) + \alpha_t \nu \tag{49}$$

Note that $\|\boldsymbol{\Sigma}_k^{t+1}\|_2 = \frac{1}{\lambda_{\min}(\boldsymbol{\Sigma}_k^{t+1\,-1})}$ and $\alpha_t = \sqrt{t+1}\alpha$, $\omega_t = 1$ , we have that

$$\|\boldsymbol{\Sigma}_k^{t+1}\|_2 \leq \frac{1}{\omega_t \lambda_{\min}\big(\boldsymbol{\Sigma}_k^{t\,-1}\big) + \alpha_t \nu} = \frac{1}{\|\boldsymbol{\Sigma}_k^t\|_2^{-1} + \sqrt{t+1}\alpha\nu} \tag{50}$$

It follows that

$$\lambda_{\min}\big(\boldsymbol{\Sigma}_k^{t+1\,-1}\big) \geq \lambda_{\min}\big(\boldsymbol{\Sigma}_k^{t\,-1}\big) + \sqrt{t+1}\alpha\nu \tag{51}$$

$$\geq \lambda_{\min}\big(\boldsymbol{\Sigma}_k^{t-1\,-1}\big) + \sqrt{t}\alpha\nu + \sqrt{t+1}\alpha\nu \tag{52}$$

$$\geq \lambda_{\min}\big(\boldsymbol{\Sigma}_k^{1\,-1}\big) + \big(\sum_{i=1}^t \sqrt{i+1}\big)\alpha\nu \tag{53}$$

$$\geq \lambda_{\min}\big(\boldsymbol{\Sigma}_k^{1\,-1}\big) + \frac{2\alpha\nu}{3}\big((t+1)^{\frac{3}{2}} - 1\big) \tag{54}$$

Note that the initialization such that $\lambda_{\min}\left(\mathbf{\Sigma}_k^{1\,-1}\right) = \|\mathbf{\Sigma}_k^1\|_2^{-1} \geq \frac{5}{3}\alpha\nu$, we have that

$$\lambda_{\min}\left(\mathbf{\Sigma}_k^{t+1\,-1}\right) \geq \frac{2\alpha\nu}{3}(t+1)^{\frac{3}{2}} + \alpha\nu = \frac{2\alpha\nu}{3}\left((t+1)^{\frac{3}{2}} + \frac{3}{2}\right) \tag{55}$$

Note that $\|\mathbf{\Sigma}_k^{t+1}\|_2 = \frac{1}{\lambda_{\min}\left(\mathbf{\Sigma}_k^{t+1\,-1}\right)}$ , we then have that

$$\|\mathbf{\Sigma}_k^{t+1}\|_2 \leq \frac{3}{2\alpha\nu}\frac{1}{(t+1)^{\frac{3}{2}} + \frac{3}{2}} \tag{56}$$

$\square$

*Proof.* (c). We now prove the upper bound of $\mathbb{E}\sum_{k=1}^K \|\mathbf{\Sigma}_k^t(\sum_{i=k}^K \hat{g}_{ik})\|_{\mathbf{\Sigma}_k^{t-1}}^2$.

Note that $\hat{g}_{ik}^t = \mathbf{\Sigma}_k^{t\,-\frac{1}{2}}z_k\big(f_i(\bar{\boldsymbol{\mu}}_i^t + \bar{\mathbf{\Sigma}}_i^{t\,\frac{1}{2}}\bar{z}_i) - f_i(\bar{\boldsymbol{\mu}}_i^t)\big)$, we have that

$$\|\mathbf{\Sigma}_k^t(\sum_{i=k}^K \hat{g}_{ik})\|_{\mathbf{\Sigma}_k^{t-1}}^2 = \|\mathbf{\Sigma}_k^t\mathbf{\Sigma}_k^{t\,-\frac{1}{2}}z_k(\sum_{i=k}^K \big(f_i(\bar{\boldsymbol{\mu}}_i^t + \bar{\mathbf{\Sigma}}_i^{t\,\frac{1}{2}}\bar{z}_i) - f_i(\bar{\boldsymbol{\mu}}_i^t))\big)\|_{\mathbf{\Sigma}_k^{t-1}}^2 \tag{57}$$

$$= \|\mathbf{\Sigma}_k^{t\,\frac{1}{2}}z_k(\sum_{i=k}^K \big(f_i(\bar{\boldsymbol{\mu}}_i^t + \bar{\mathbf{\Sigma}}_i^{t\,\frac{1}{2}}\bar{z}_i) - f_i(\bar{\boldsymbol{\mu}}_i^t))\big)\|_{\mathbf{\Sigma}_k^{t-1}}^2 \tag{58}$$

$$= \|z_k(\sum_{i=k}^K \big(f_i(\bar{\boldsymbol{\mu}}_i^t + \bar{\mathbf{\Sigma}}_i^{t\,\frac{1}{2}}\bar{z}_i) - f_i(\bar{\boldsymbol{\mu}}_i^t))\big)\|_2^2 \tag{59}$$

$$= \big(\sum_{i=k}^K \big(f_i(\bar{\boldsymbol{\mu}}_i^t + \bar{\mathbf{\Sigma}}_i^{t\,\frac{1}{2}}\bar{z}_i) - f_i(\bar{\boldsymbol{\mu}}_i^t))\big)^2\|z_k\|_2^2 \tag{60}$$

Note that $f_i(\bar{x})$ is $L_i$-Lipschitz continuous function, we then have that

$$\big(\sum_{i=k}^K \big(f_i(\bar{\boldsymbol{\mu}}_i^t + \bar{\mathbf{\Sigma}}_i^{t\,\frac{1}{2}}\bar{z}_i) - f_i(\bar{\boldsymbol{\mu}}_i^t))\big)^2 \leq (K-k+1)(\sum_{i=k}^K \big(f_i(\bar{\boldsymbol{\mu}}_i^t + \bar{\mathbf{\Sigma}}_i^{t\,\frac{1}{2}}\bar{z}_i) - f_i(\bar{\boldsymbol{\mu}}_i^t))^2) \tag{61}$$

$$\leq (K-k+1)(\sum_{i=k}^K L_i^2\|\bar{\boldsymbol{\mu}}_i^t + \bar{\mathbf{\Sigma}}_i^{t\,\frac{1}{2}}\bar{z}_i - \bar{\boldsymbol{\mu}}_i^t\|_2^2) \tag{62}$$

$$\leq (K-k+1)(\sum_{i=k}^K L_i^2\|\bar{\mathbf{\Sigma}}_i^{t\,\frac{1}{2}}\|_2^2\|\bar{z}_i\|_2^2) \tag{63}$$

$$= (K-k+1)(\sum_{i=k}^K L_i^2\|\bar{\mathbf{\Sigma}}_i^t\|_2\|\bar{z}_i\|_2^2) \tag{64}$$

Plug Eq.(64) into Eq.(60), we have that

$$\|\mathbf{\Sigma}_k^t(\sum_{i=k}^K \hat{g}_{ik})\|_{\mathbf{\Sigma}_k^{t-1}}^2 \leq (K-k+1)\|z_k\|_2^2(\sum_{i=k}^K L_i^2\|\bar{\mathbf{\Sigma}}_i^t\|_2\|\bar{z}_i\|_2^2) \tag{65}$$

It follows that

$$\sum_{k=1}^{K} \|\mathbf{\Sigma}_k^t (\sum_{i=k}^{K} \hat{g}_{ik})\|_{\mathbf{\Sigma}_k^{t-1}}^2 \leq \sum_{k=1}^{K} (K-k+1)\|\mathbf{z}_k\|_2^2 (\sum_{i=k}^{K} L_i^2 \|\bar{\mathbf{\Sigma}}_i^t\|_2 \|\bar{\mathbf{z}}_i\|_2^2) \tag{66}$$

$$= \sum_{i=1}^{K} \sum_{k=1}^{i} \left( (K-k+1)\|\mathbf{z}_k\|_2^2 L_i^2 \|\bar{\mathbf{\Sigma}}_i^t\|_2 \|\bar{\mathbf{z}}_i\|_2^2 \right) \tag{67}$$

$$= \sum_{i=1}^{K} L_i^2 \|\bar{\mathbf{\Sigma}}_i^t\|_2 \|\bar{\mathbf{z}}_i\|_2^2 \left( \sum_{k=1}^{i} (K-k+1)\|\mathbf{z}_k\|_2^2 \right) \tag{68}$$

$$\leq \sum_{i=1}^{K} L_i^2 \|\bar{\mathbf{\Sigma}}_i^t\|_2 \|\bar{\mathbf{z}}_i\|_2^2 K \|\bar{\mathbf{z}}_i\|_2^2 \tag{69}$$

$$= \sum_{i=1}^{K} K L_i^2 \|\bar{\mathbf{\Sigma}}_i^t\|_2 \|\bar{\mathbf{z}}_i\|_2^4 \tag{70}$$

In addition, note that for $z \sim \mathcal{N}(0, \sigma^2)$, we have $\mathbb{E}[z^4] = 3\sigma^4$. It follows that

$$\mathbb{E}\|\bar{\mathbf{z}}_i\|_2^4 = \sum_{j=1}^{id} \mathbb{E}[z_j^4] + \sum_{j_1=1}^{id} \sum_{j_2 \neq j_1}^{id} \mathbb{E}[z_{j_1}^2 z_{j_2}^2] = 3id + id(id-1) = i^2 d^2 + 2id \tag{71}$$

We then have that

$$\mathbb{E}\sum_{k=1}^{K} \|\mathbf{\Sigma}_k^t (\sum_{i=k}^{K} \hat{g}_{ik})\|_{\mathbf{\Sigma}_k^{t-1}}^2 \leq \sum_{i=1}^{K} K L_i^2 \|\bar{\mathbf{\Sigma}}_i^t\|_2 \mathbb{E}\|\bar{\mathbf{z}}_i\|_2^4 \tag{72}$$

$$\leq \sum_{i=1}^{K} K L_i^2 \|\bar{\mathbf{\Sigma}}_i^t\|_2 (id+1)^2 \tag{73}$$

Note that $\bar{\mathbf{\Sigma}}_i^t = diag(\mathbf{\Sigma}_1^t, \cdots, \mathbf{\Sigma}_i^t)$ is a block-wise diagonal matrix, we have $\|\bar{\mathbf{\Sigma}}_i^t\|_2 \leq \max_{k \in \{1, \cdots, i\}} \|\mathbf{\Sigma}_k^t\|_2$. Then we know that

$$\mathbb{E}\sum_{k=1}^{K} \|\mathbf{\Sigma}_k^t (\sum_{i=k}^{K} \hat{g}_{ik})\|_{\mathbf{\Sigma}_k^{t-1}}^2 \leq \sum_{i=1}^{K} K L_i^2 \|\bar{\mathbf{\Sigma}}_i^t\|_2 (id+1)^2 \tag{74}$$

$$\leq \max_{k \in \{1, \cdots, K\}} \|\mathbf{\Sigma}_k^t\|_2 \sum_{i=1}^{K} K L_i^2 (id+1)^2 \tag{75}$$

From Lemma C.2 (b), we know that $\|\mathbf{\Sigma}_k^t\|_2 \leq \frac{3}{2\alpha\nu} \frac{1}{t^{\frac{3}{2}}}$ for $\forall k \in \{1, \cdots, K\}$. Then, we have

$$\mathbb{E}\sum_{k=1}^{K} \|\mathbf{\Sigma}_k^t (\sum_{i=k}^{K} \hat{g}_{ik})\|_{\mathbf{\Sigma}_k^{t-1}}^2 \leq \max_{k \in \{1, \cdots, K\}} \|\mathbf{\Sigma}_k^t\|_2 \sum_{i=1}^{K} K L_i^2 (id+1)^2 \tag{76}$$

$$\leq \frac{3 \sum_{i=1}^{K} K L_i^2 (id+1)^2}{2 t^{\frac{3}{2}} \alpha\nu} \tag{77}$$

Note that the square norm $\|\cdot\|^2_{\Sigma_k^{t-1}}$ is a convex function, then for the average of $N$ i.i.d. sampled gradient estimator $\hat{g}_{ik}^j, j \in \{1, \cdots, N\}$, we have

$$\mathbb{E}\sum_{k=1}^{K}\|\boldsymbol{\Sigma}_k^t(\sum_{i=k}^{K}\frac{1}{N}\sum_{j=1}^{N}\hat{g}_{ik}^j)\|^2_{\boldsymbol{\Sigma}_k^{t-1}} \leq \frac{1}{N}\sum_{j=1}^{N}\mathbb{E}\sum_{k=1}^{K}\|\boldsymbol{\Sigma}_k^t(\sum_{i=k}^{K}\hat{g}_{ik}^j)\|^2_{\boldsymbol{\Sigma}_k^{t-1}} \tag{78}$$

$$= \frac{N}{N}\mathbb{E}\sum_{k=1}^{K}\|\boldsymbol{\Sigma}_k^t(\sum_{i=k}^{K}\hat{g}_{ik})\|^2_{\boldsymbol{\Sigma}_k^{t-1}} \tag{79}$$

$$\leq \frac{3\sum_{i=1}^{K}KL_i^2(id+1)^2}{2t^{\frac{3}{2}}\alpha\nu} \tag{80}$$

$\square$

**Lemma C.3.** *Denote $G_i^t = \nabla_{\bar{\boldsymbol{\Sigma}}_i=\bar{\boldsymbol{\Sigma}}_i^t}J_i(\bar{\boldsymbol{\mu}}_i^t, \bar{\boldsymbol{\Sigma}}_i^t)$. Suppose assumption 5.2 holds, using the parameter setting in Theorem 5.4, and the covariance matrix update $\hat{G}_k^t = \boldsymbol{\Sigma}_k^{t\ -\frac{1}{2}}\boldsymbol{H}_k^t\boldsymbol{\Sigma}_k^{t\ -\frac{1}{2}}$ are positive semidefinite matrix that satisfies $\nu\boldsymbol{I} \preceq \hat{G}_k^t$. Apply the update rule $\boldsymbol{\Sigma}_k^{t+1\ -1} = \omega_t\boldsymbol{\Sigma}_k^{t\ -1} + \alpha_t\hat{G}_k^t$, for $k \in \{1, \cdots, K\}$. Then we have*

$$tr(G_i^t\bar{\boldsymbol{\Sigma}}_i^t) \leq |tr(G_i^t\bar{\boldsymbol{\Sigma}}_i^t)| \leq \frac{L_i id}{2t^{\frac{3}{4}}}\sqrt{\frac{3}{\alpha\nu}} \tag{81}$$

*Proof.*

$$tr(G_i^t\bar{\boldsymbol{\Sigma}}_i^t) = tr(\bar{\boldsymbol{\Sigma}}_i^{t\frac{1}{2}}G_i^t\bar{\boldsymbol{\Sigma}}_i^{t\frac{1}{2}}) \tag{82}$$

$$= \frac{1}{2}tr\left(\bar{\boldsymbol{\Sigma}}_i^{t\frac{1}{2}}\mathbb{E}_{\mathcal{N}(\bar{\boldsymbol{\mu}}_i^t,\bar{\boldsymbol{\Sigma}}_i^t)}\left[(\bar{\boldsymbol{\Sigma}}_i^{t\ -1}(\bar{\boldsymbol{x}}_i-\bar{\boldsymbol{\mu}}_i^t)(\bar{\boldsymbol{x}}_i-\bar{\boldsymbol{\mu}}_i^t)^\top\bar{\boldsymbol{\Sigma}}_i^{t\ -1}-\bar{\boldsymbol{\Sigma}}_i^{t\ -1})f_i(\bar{\boldsymbol{x}}_i)\right]\bar{\boldsymbol{\Sigma}}_i^{t\frac{1}{2}}\right) \tag{83}$$

$$= \frac{1}{2}tr\left(\mathbb{E}_{\mathcal{N}(\bar{\boldsymbol{\mu}}_i^t,\bar{\boldsymbol{\Sigma}}_i^t)}\left[(\bar{\boldsymbol{\Sigma}}_i^{t\ -\frac{1}{2}}(\bar{\boldsymbol{x}}_i-\bar{\boldsymbol{\mu}}_i^t)(\bar{\boldsymbol{x}}_i-\bar{\boldsymbol{\mu}}_i^t)^\top\bar{\boldsymbol{\Sigma}}_i^{t\ -\frac{1}{2}}-\boldsymbol{I})f_i(\bar{\boldsymbol{x}}_i)\right]\right) \tag{84}$$

$$= \frac{1}{2}tr\left(\mathbb{E}_{\bar{\boldsymbol{z}}\sim\mathcal{N}(\boldsymbol{0},\boldsymbol{I})}\left[(\bar{\boldsymbol{z}}\bar{\boldsymbol{z}}^\top-\boldsymbol{I})f_i(\bar{\boldsymbol{\mu}}_i^t+\bar{\boldsymbol{\Sigma}}_i^{t\frac{1}{2}}\bar{\boldsymbol{z}})\right]\right) \tag{85}$$

$$= \frac{1}{2}tr\left(\mathbb{E}_{\bar{\boldsymbol{z}}\sim\mathcal{N}(\boldsymbol{0},\boldsymbol{I})}\left[(\bar{\boldsymbol{z}}\bar{\boldsymbol{z}}^\top-\boldsymbol{I})(f_i(\bar{\boldsymbol{\mu}}_i^t+\bar{\boldsymbol{\Sigma}}_i^{t\frac{1}{2}}\bar{\boldsymbol{z}})-f_i(\bar{\boldsymbol{\mu}}_i^t))\right]\right) \tag{86}$$

$$= \frac{1}{2}\mathbb{E}_{\bar{\boldsymbol{z}}\sim\mathcal{N}(\boldsymbol{0},\boldsymbol{I})}\left[(\sum_{j=1}^{id}(z_j^2-1))(f_i(\bar{\boldsymbol{\mu}}_i^t+\bar{\boldsymbol{\Sigma}}_i^{t\frac{1}{2}}\bar{\boldsymbol{z}})-f_i(\bar{\boldsymbol{\mu}}_i^t))\right] \tag{87}$$

where $z_j$ denotes the $j^{th}$ element in $\bar{\boldsymbol{z}}$.

From Cauchy–Schwarz inequality $|\mathbb{E}[XY]| \leq \sqrt{\mathbb{E}[X^2]\mathbb{E}[Y^2]}$, we know that

$$|tr(G_i^t\bar{\boldsymbol{\Sigma}}_i^t)| = \frac{1}{2}|\mathbb{E}_{\bar{\boldsymbol{z}}\sim\mathcal{N}(\boldsymbol{0},\boldsymbol{I})}\left[(\sum_{j=1}^{id}(z_j^2-1))(f_i(\bar{\boldsymbol{\mu}}_i^t+\bar{\boldsymbol{\Sigma}}_i^{t\frac{1}{2}}\bar{\boldsymbol{z}})-f_i(\bar{\boldsymbol{\mu}}_i^t))\right]| \tag{88}$$

$$\leq \frac{1}{2}\sqrt{\mathbb{E}_{\bar{\boldsymbol{z}}\sim\mathcal{N}(\boldsymbol{0},\boldsymbol{I})}\left[(\sum_{j=1}^{id}(z_j^2-1))^2\right]\mathbb{E}_{\bar{\boldsymbol{z}}\sim\mathcal{N}(\boldsymbol{0},\boldsymbol{I})}\left[(f_i(\bar{\boldsymbol{\mu}}_i^t+\bar{\boldsymbol{\Sigma}}_i^{t\frac{1}{2}}\bar{\boldsymbol{z}})-f_i(\bar{\boldsymbol{\mu}}_i^t))^2\right]} \tag{89}$$

We first check the term $\mathbb{E}_{\bar{z} \sim \mathcal{N}(\mathbf{0}, \mathbf{I})}\big[(\sum_{j=1}^{id}(z_j^2 - 1))^2\big]$.

$$\mathbb{E}_{\bar{z} \sim \mathcal{N}(\mathbf{0}, \mathbf{I})}\big[(\sum_{j=1}^{id}(z_j^2 - 1))^2\big] = \sum_{j=1}^{id} \mathbb{E}(z_j^2 - 1)^2 + \sum_{j_1=1}^{id} \sum_{j_2 \neq j_1}^{id} \mathbb{E}(z_{j_1}^2 - 1)(z_{j_2}^2 - 1) \tag{90}$$

$$= \sum_{j=1}^{id} \mathbb{E}(z_j^4 - 2z_j^2 + 1) + \sum_{j_1=1}^{id} \sum_{j_2 \neq j_1}^{id} \mathbb{E}(z_{j_1}^2 - 1)\mathbb{E}(z_{j_2}^2 - 1) \tag{91}$$

$$= \sum_{j=1}^{id} \mathbb{E}(z_j^4 - 2z_j^2 + 1) = \sum_{j=1}^{id}[3 - 2 + 1] = 2id \tag{92}$$

We now check the term $\mathbb{E}_{\bar{z} \sim \mathcal{N}(\mathbf{0}, \mathbf{I})}\big[(f_i(\bar{\boldsymbol{\mu}}_i^t + \bar{\boldsymbol{\Sigma}}_i^{t\frac{1}{2}}\bar{z}) - f_i(\bar{\boldsymbol{\mu}}_i^t))^2\big]$. Note that $f_i(\boldsymbol{x})$ is $L_i$-Lipschitz continuous function, we then have that

$$\mathbb{E}_{\bar{z} \sim \mathcal{N}(\mathbf{0}, \mathbf{I})}\big[(f_i(\bar{\boldsymbol{\mu}}_i^t + \bar{\boldsymbol{\Sigma}}_i^{t\frac{1}{2}}\bar{z}) - f_i(\bar{\boldsymbol{\mu}}_i^t))^2\big] \leq L_i^2 \mathbb{E}_{\bar{z} \sim \mathcal{N}(\mathbf{0}, \mathbf{I})}\big[\|\bar{\boldsymbol{\Sigma}}_i^{t\frac{1}{2}}\bar{z}\|_2^2\big] \tag{93}$$

$$\leq L_i^2 \|\bar{\boldsymbol{\Sigma}}_i^{t\frac{1}{2}}\|_2^2 \mathbb{E}_{\bar{z} \sim \mathcal{N}(\mathbf{0}, \mathbf{I})}\|\bar{z}\|_2^2 \tag{94}$$

$$= L_i^2 \|\bar{\boldsymbol{\Sigma}}_i^t\|_2 id \tag{95}$$

From Lemma C.2 (b) we know that

$$\|\bar{\boldsymbol{\Sigma}}_i^t\|_2 = \max_{k \in \{1, \cdots, i\}} \|\boldsymbol{\Sigma}_k^t\|_2 \leq \frac{3}{2\alpha\nu} \frac{1}{t^{\frac{3}{2}}} \tag{96}$$

Together with Eq.(95) and Eq.(96 ), we know that

$$\mathbb{E}_{\bar{z} \sim \mathcal{N}(\mathbf{0}, \mathbf{I})}\big[(f_i(\bar{\boldsymbol{\mu}}_i^t + \bar{\boldsymbol{\Sigma}}_i^{t\frac{1}{2}}\bar{z}) - f_i(\bar{\boldsymbol{\mu}}_i^t))^2\big] \leq \frac{3L_i^2 id}{2t^{\frac{3}{2}}\alpha\nu} \tag{97}$$

Plug Eq.(97) and Eq.(92) into Eq.(89), we have that

$$|\text{tr}(G_i^t \bar{\boldsymbol{\Sigma}}_i^t)| \leq \frac{L_i id}{2t^{\frac{3}{4}}} \sqrt{\frac{3}{\alpha\nu}} \tag{98}$$

$\square$

**Lemma C.4.** *Given a convex function $f(x)$, for Gaussian distribution with parameters $\boldsymbol{\theta} := \{\boldsymbol{\mu}, \Sigma^{\frac{1}{2}}\}$, let $\bar{J}(\boldsymbol{\theta}) := \mathbb{E}_{p(\boldsymbol{x};\boldsymbol{\theta})}[f(\boldsymbol{x})]$. Then $\bar{J}(\boldsymbol{\theta})$ is a convex function with respect to $\boldsymbol{\theta}$.*

*Proof.* For $\lambda \in [0, 1]$, we have

$$\lambda\bar{J}(\boldsymbol{\theta}_1) + (1 - \lambda)\bar{J}(\boldsymbol{\theta}_2) = \lambda\mathbb{E}_{\boldsymbol{z} \sim \mathcal{N}(\mathbf{0}, \mathbf{I})}[f(\boldsymbol{\mu}_1 + \Sigma_1^{\frac{1}{2}}\boldsymbol{z})] + (1 - \lambda)\mathbb{E}_{\boldsymbol{z} \sim \mathcal{N}(\mathbf{0}, \mathbf{I})}[f(\boldsymbol{\mu}_2 + \Sigma_2^{\frac{1}{2}}\boldsymbol{z})] \tag{99}$$

$$= \mathbb{E}[\lambda f(\boldsymbol{\mu}_1 + \Sigma_1^{\frac{1}{2}}\boldsymbol{z}) + (1 - \lambda)f(\boldsymbol{\mu}_2 + \Sigma_2^{\frac{1}{2}}\boldsymbol{z})] \tag{100}$$

$$\geq \mathbb{E}[f\left(\lambda\boldsymbol{\mu}_1 + (1 - \lambda)\boldsymbol{\mu}_2 + (\lambda\Sigma_1^{\frac{1}{2}} + (1 - \lambda)\Sigma_2^{\frac{1}{2}})\boldsymbol{z}\right)] \tag{101}$$

$$= \bar{J}(\lambda\boldsymbol{\theta}_1 + (1 - \lambda)\boldsymbol{\theta}_2) \tag{102}$$

$\square$

**Lemma C.5.** *Given a convex function $f(x)$, let $J(\boldsymbol{\mu}, \boldsymbol{\Sigma}) := \mathbb{E}_{\boldsymbol{x} \sim \mathcal{N}(\boldsymbol{\mu}, \boldsymbol{\Sigma})}[f(\boldsymbol{x})]$. Then, we have*

$$f(\boldsymbol{\mu}) - f(\boldsymbol{\mu}^*) \leq J(\boldsymbol{\mu}, \boldsymbol{\Sigma}) - J(\boldsymbol{\mu}^*, \mathbf{0}) \tag{103}$$

*Proof.* From the definition of $J(\boldsymbol{\mu}, \boldsymbol{\Sigma})$, we know that $f(\boldsymbol{\mu}^*) = J(\boldsymbol{\mu}^*, \mathbf{0})$.

Note that $f(\boldsymbol{x})$ is a convex function, we have that

$$f(\boldsymbol{\mu}) = f(\mathbb{E}_{\boldsymbol{x} \sim \mathcal{N}(\boldsymbol{\mu}, \boldsymbol{\Sigma})}[\boldsymbol{x}]) \leq \mathbb{E}_{\boldsymbol{x} \sim \mathcal{N}(\boldsymbol{\mu}, \boldsymbol{\Sigma})}[f(\boldsymbol{x})] = J(\boldsymbol{\mu}, \boldsymbol{\Sigma}) \tag{104}$$

It follows that

$$f(\boldsymbol{\mu}) - f(\boldsymbol{\mu}^*) \leq J(\boldsymbol{\mu}, \boldsymbol{\Sigma}) - J(\boldsymbol{\mu}^*, \mathbf{0}) \tag{105}$$

$\square$

# D  PROOF OF THEOREM 5.4

In this section, we prove our main Theorem 5.4. We decompose the proof into two parts. The proof of Theorem D.1 and the proof of Theorem D.2. Together with Theorem D.1 and the Theorem D.2, we achieve our main Theorem 5.4.

**Theorem D.1.** *Suppose the assumptions 5.1 5.2 5.3 holds. Set $\beta_t = t\beta$ with $\beta > 0$, $\alpha_t = \sqrt{t+1}\alpha$ with $\alpha > 0$, and $\gamma_t = \frac{\alpha\nu}{\beta\sqrt{t+1}}$, and $\nu > 0$, and $\omega_t = 1$. Initialize $\boldsymbol{\Sigma}_k^1$ such that $\|\boldsymbol{\Sigma}_k^1\|_2^{-1} \geq \frac{5}{3}\alpha\nu$ for $\forall k \in \{1, \cdots, K\}$. Suppose the constraints $\boldsymbol{H}_k^t \preceq \frac{1}{\alpha_t}(\frac{\beta_{t+1}}{\beta_t} - \omega_t)\boldsymbol{I} + \frac{\beta_{t+1}\gamma_t}{\alpha_t}\boldsymbol{\Sigma}_k^t$ and $\nu\boldsymbol{I} \preceq \hat{G}_k^t = \boldsymbol{\Sigma}_k^{t\,-\frac{1}{2}}\boldsymbol{H}_k^t\boldsymbol{\Sigma}_k^{t\,-\frac{1}{2}}$ always have feasible solutions. Then, running Algorithm 2 with $T$-steps, we have*

$$\frac{1}{T}\sum_{t=1}^{T}\sum_{k=1}^{K} f_k(\bar{\boldsymbol{\mu}}_k^t) - \sum_{k=1}^{K} f_k(\bar{\boldsymbol{\mu}}_k^*) \leq \frac{\sum_{k=1}^{K}\|\boldsymbol{\mu}_k^1 - \boldsymbol{\mu}_k^*\|_{\boldsymbol{\Sigma}_k^{1-1}}^2}{2\beta T} + \frac{2\sqrt{T+1}C_1}{T} + \frac{4(T+1)^{\frac{1}{4}}C_2}{T} + \frac{\sqrt{T+2}C_3}{T}$$

$$(106)$$

$$\leq O(\frac{d^2 K^4}{\sqrt{T}})$$

$$(107)$$

*where $\bar{\boldsymbol{\mu}}_k^t = [\boldsymbol{\mu}_1^{t\top}, \cdots, \boldsymbol{\mu}_k^{t\top}]^\top$ and $\bar{\boldsymbol{\mu}}_k^* = [\boldsymbol{\mu}_1^{*\top}, \cdots, \boldsymbol{\mu}_k^{*\top}]^\top$. And $C_1 = \frac{3\beta\sum_{i=1}^{K} KL_i^2(id+1)^2}{2\alpha\nu}$ and $C_2 = \frac{\sum_{i=1}^{K}\sqrt{3}idL_i}{\sqrt{\alpha\nu}}$, $C_3 = \frac{\alpha\nu B}{\beta}$*

*Proof.* For $\forall k \in \{1, \cdots, K\}$, we have

$$\|\boldsymbol{\mu}_k^{t+1} - \boldsymbol{\mu}_k^*\|_{\boldsymbol{\Sigma}_k^{t-1}}^2$$

$$= \|\boldsymbol{\mu}_k^t - \beta_t\,\boldsymbol{\Sigma}_k^t((\sum_{i=k}^{K}\hat{g}_{ik}^t) + \gamma_t\boldsymbol{\mu}_k^t) - \boldsymbol{\mu}_k^*\|_{\boldsymbol{\Sigma}_k^{t-1}}^2$$

$$(108)$$

$$= \|\boldsymbol{\mu}_k^t - \boldsymbol{\mu}_k^*\|_{\boldsymbol{\Sigma}_k^{t-1}}^2 - 2\beta_t\left\langle\boldsymbol{\Sigma}_k^t((\sum_{i=k}^{K}\hat{g}_{ik}^t) + \gamma_t\boldsymbol{\mu}_k^t), \boldsymbol{\mu}_k^t - \boldsymbol{\mu}_k^*\right\rangle_{\boldsymbol{\Sigma}_k^{t-1}} + \beta_t^2\|\boldsymbol{\Sigma}_k^t((\sum_{i=k}^{K}\hat{g}_{ik}^t) + \gamma_t\boldsymbol{\mu}_k^t)\|_{\boldsymbol{\Sigma}_k^{t-1}}^2$$

$$(109)$$

$$= \|\boldsymbol{\mu}_k^t - \boldsymbol{\mu}_k^*\|_{\boldsymbol{\Sigma}_k^{t-1}}^2 - 2\beta_t\left\langle\gamma_t\boldsymbol{\mu}_k^t + \sum_{i=k}^{K}\hat{g}_{ik}^t, \boldsymbol{\mu}_k^t - \boldsymbol{\mu}_k^*\right\rangle + \beta_t^2\|\boldsymbol{\Sigma}_k^t((\sum_{i=k}^{K}\hat{g}_{ik}^t) + \gamma_t\boldsymbol{\mu}_k^t)\|_{\boldsymbol{\Sigma}_k^{t-1}}^2$$

$$(110)$$

Note that

$$\gamma_t\langle\boldsymbol{\mu}_k^t, \boldsymbol{\mu}_k^t - \boldsymbol{\mu}_k^*\rangle = \frac{\gamma_t}{2}\|\boldsymbol{\mu}_k^t - \boldsymbol{\mu}_k^*\|_2^2 - \frac{\gamma_t}{2}\|\boldsymbol{\mu}_k^*\|_2^2 + \frac{\gamma_t}{2}\|\boldsymbol{\mu}_k^t\|_2^2$$

$$(111)$$

Plug Eq.(111) into Eq.(110), we have that

$$\|\boldsymbol{\mu}_k^{t+1} - \boldsymbol{\mu}_k^*\|_{\boldsymbol{\Sigma}_k^{t-1}}^2$$

$$= \|\boldsymbol{\mu}_k^t - \boldsymbol{\mu}_k^*\|_{\boldsymbol{\Sigma}_k^{t-1}}^2 - 2\beta_t\langle\sum_{i=k}^{K}\hat{g}_{ik}^t, \boldsymbol{\mu}_k^t - \boldsymbol{\mu}_k^*\rangle - \beta_t\gamma_t(\|\boldsymbol{\mu}_k^t - \boldsymbol{\mu}_k^*\|_2^2 - \|\boldsymbol{\mu}_k^*\|_2^2 + \|\boldsymbol{\mu}_k^t\|_2^2) + \beta_t^2\|\boldsymbol{\Sigma}_k^t((\sum_{i=k}^{K}\hat{g}_{ik}^t) + \gamma_t\boldsymbol{\mu}_k^t)\|_{\boldsymbol{\Sigma}_k^{t-1}}^2$$

$$(112)$$

From Lemma C.1, we then have that

$$\|\boldsymbol{\mu}_k^{t+1} - \boldsymbol{\mu}_k^*\|_{\boldsymbol{\Sigma}_k^{t-1}}^2 \leq \|\boldsymbol{\mu}_k^t - \boldsymbol{\mu}_k^*\|_{\boldsymbol{\Sigma}_k^{t-1}}^2 - 2\beta_t\langle\sum_{i=k}^{K}\hat{g}_{ik}^t, \boldsymbol{\mu}_k^t - \boldsymbol{\mu}_k^*\rangle - \beta_t\gamma_t(\|\boldsymbol{\mu}_k^t - \boldsymbol{\mu}_k^*\|_2^2 - \|\boldsymbol{\mu}_k^*\|_2^2 + \|\boldsymbol{\mu}_k^t\|_2^2)$$

$$+ 2\beta_t^2\|\boldsymbol{\Sigma}_k^t((\sum_{i=k}^{K}\hat{g}_{ik}^t)\|_{\boldsymbol{\Sigma}_k^{t-1}}^2 + 2\beta_t^2\|\gamma_t\boldsymbol{\Sigma}_k^{t\,\frac{1}{2}}\boldsymbol{\mu}_k^t\|_2^2$$

$$(113)$$

From Lemma C.2 (b), we know that $\|\boldsymbol{\Sigma}_k^t\|_2 \leq \frac{3}{2\alpha\nu}\frac{1}{t\sqrt{t}+3/2}$, together with the setting $\beta_t = t\beta$ and $\gamma_t = \frac{\alpha\nu}{\beta\sqrt{t+1}}$, we know that

$$-\gamma_t\|\boldsymbol{\mu}_k^t\|_2^2 + 2\beta_t\|\gamma_t\boldsymbol{\Sigma}_k^{t\frac{1}{2}}\boldsymbol{\mu}_k^t\|_2^2 = -\gamma_t\|\boldsymbol{\mu}_k^t\|_2^2 + 2\beta_t\gamma_t^2\|\boldsymbol{\Sigma}_k^{t\frac{1}{2}}\boldsymbol{\mu}_k^t\|_2^2 \tag{114}$$

$$\leq -\gamma_t\|\boldsymbol{\mu}_k^t\|_2^2 + 2\beta_t\gamma_t^2\|\boldsymbol{\Sigma}_k^{t\frac{1}{2}}\|_2^2\|\boldsymbol{\mu}_k^t\|_2^2 \tag{115}$$

$$= -\gamma_t\|\boldsymbol{\mu}_k^t\|_2^2 + 2\beta_t\gamma_t^2\|\boldsymbol{\Sigma}_k^t\|_2\|\boldsymbol{\mu}_k^t\|_2^2 \tag{116}$$

$$= \gamma_t\|\boldsymbol{\mu}_k^t\|_2^2(-1 + 2\beta_t\gamma_t\|\boldsymbol{\Sigma}_k^t\|_2) \tag{117}$$

$$\leq \gamma_t\|\boldsymbol{\mu}_k^t\|_2^2(-1 + 2t\beta\frac{\alpha\nu}{\beta\sqrt{t+1}}\frac{3}{2\alpha\nu}\frac{1}{3/2+t\sqrt{t}}) \tag{118}$$

$$= \gamma_t\|\boldsymbol{\mu}_k^t\|_2^2(-1 + \frac{3t}{\frac{3}{2}\sqrt{(t+1)}+t\sqrt{t(t+1)}}) \tag{119}$$

We now check the term $(-1 + \frac{3t}{\frac{3}{2}\sqrt{(t+1)}+t\sqrt{t(t+1)}})$. For $t=1$ and $t=2$, it is easy to see the term $(-1 + \frac{3t}{\frac{3}{2}\sqrt{(t+1)}+t\sqrt{t(t+1)}}) \leq 0$. For $t \geq 3$, we have that

$$\frac{3}{2}\sqrt{(t+1)} + t\sqrt{t(t+1)} - 3t \geq \frac{3}{2}\sqrt{(t+1)} + t^2 - 3t \geq 0 \tag{120}$$

It follows that $(-1 + \frac{3t}{\frac{3}{2}\sqrt{(t+1)}+t\sqrt{t(t+1)}}) \leq 0$. Thus, we have that

$$-\gamma_t\|\boldsymbol{\mu}_k^t\|_2^2 + 2\beta_t\|\gamma_t\boldsymbol{\Sigma}_k^{t\frac{1}{2}}\boldsymbol{\mu}_k^t\|_2^2 \leq 0 \tag{121}$$

Plug the inequality (121) into inequality (113), we know that

$$\|\boldsymbol{\mu}_k^{t+1} - \boldsymbol{\mu}_k^*\|_{\boldsymbol{\Sigma}_k^{t-1}}^2 \leq \|\boldsymbol{\mu}_k^t - \boldsymbol{\mu}_k^*\|_{\boldsymbol{\Sigma}_k^{t-1}}^2 - 2\beta_t\langle\sum_{i=k}^K \hat{g}_{ik}^t, \boldsymbol{\mu}_k^t - \boldsymbol{\mu}_k^*\rangle - \beta_t\gamma_t(\|\boldsymbol{\mu}_k^t-\boldsymbol{\mu}_k^*\|_2^2-\|\boldsymbol{\mu}_k^*\|_2^2)$$

$$+ 2\beta_t^2\|\boldsymbol{\Sigma}_k^t((\sum_{i=k}^K \hat{g}_{ik}^t))\|_{\boldsymbol{\Sigma}_k^{t-1}}^2 \tag{122}$$

It follows that

$$\sum_{k=1}^K\|\boldsymbol{\mu}_k^{t+1} - \boldsymbol{\mu}_k^*\|_{\boldsymbol{\Sigma}_k^{t-1}}^2 \leq \sum_{k=1}^K\|\boldsymbol{\mu}_k^t - \boldsymbol{\mu}_k^*\|_{\boldsymbol{\Sigma}_k^{t-1}}^2 - 2\beta_t\sum_{k=1}^K\langle\sum_{i=k}^K \hat{g}_{ik}^t, \boldsymbol{\mu}_k^t - \boldsymbol{\mu}_k^*\rangle + 2\beta_t^2\sum_{k=1}^K\|\boldsymbol{\Sigma}_k^t(\sum_{i=k}^K \hat{g}_{ik})\|_{\boldsymbol{\Sigma}_k^{t-1}}^2$$

$$- \sum_{k=1}^K\beta_t\gamma_t(\|\boldsymbol{\mu}_k^t-\boldsymbol{\mu}_k^*\|_2^2-\|\boldsymbol{\mu}_k^*\|_2^2) \tag{123}$$

Then, we have that

$$\mathbb{E}\sum_{k=1}^K\|\boldsymbol{\mu}_k^{t+1} - \boldsymbol{\mu}_k^*\|_{\boldsymbol{\Sigma}_k^{t-1}}^2 \leq \sum_{k=1}^K\mathbb{E}\|\boldsymbol{\mu}_k^t - \boldsymbol{\mu}_k^*\|_{\boldsymbol{\Sigma}_k^{t-1}}^2 - 2\beta_t\mathbb{E}\sum_{k=1}^K\langle\sum_{i=k}^K \hat{g}_{ik}^t, \boldsymbol{\mu}_k^t - \boldsymbol{\mu}_k^*\rangle + 2\beta_t^2\mathbb{E}\sum_{k=1}^K\|\boldsymbol{\Sigma}_k^t(\sum_{i=k}^K \hat{g}_{ik})\|_{\boldsymbol{\Sigma}_k^{t-1}}^2$$

$$- \sum_{k=1}^K\beta_t\gamma_t(\mathbb{E}\|\boldsymbol{\mu}_k^t-\boldsymbol{\mu}_k^*\|_2^2-\|\boldsymbol{\mu}_k^*\|_2^2) \tag{124}$$

Note that $\bar{\boldsymbol{\mu}}_k^t = [\boldsymbol{\mu}_1^{t\top}, \cdots, \boldsymbol{\mu}_k^{t\top}]^\top$ and $\bar{\boldsymbol{\mu}}_k^* = [\boldsymbol{\mu}_1^{*\top}, \cdots, \boldsymbol{\mu}_k^{*\top}]^\top$, together with Lemma C.2 (a), we have that

$$\mathbb{E}\sum_{k=1}^K\langle\sum_{i=k}^K \hat{g}_{ik}^t, \boldsymbol{\mu}_k^t - \boldsymbol{\mu}_k^*\rangle = \mathbb{E}\sum_{i=1}^K\langle\sum_{k=1}^i \hat{g}_{ik}^t, \boldsymbol{\mu}_k^t - \boldsymbol{\mu}_k^*\rangle = \sum_{i=1}^K\langle g_i^t, \bar{\boldsymbol{\mu}}_i^t - \bar{\boldsymbol{\mu}}_i^*\rangle \tag{125}$$

where $g_i^t = \nabla_{\bar{\boldsymbol{\mu}}_i^t}\mathbb{E}_{\boldsymbol{x}\sim\mathcal{N}(\bar{\boldsymbol{\mu}}_i^t, \bar{\boldsymbol{\Sigma}}_i^t)}[f_i(\boldsymbol{x})] = \nabla_{\bar{\boldsymbol{\mu}}_i^t}J_i(\bar{\boldsymbol{\mu}}_i^t, \bar{\boldsymbol{\Sigma}}_i^t)$

From Eq. (125) and Eq. (124), we have that

$$\sum_{i=1}^{K} \langle g_i^t, \bar{\boldsymbol{\mu}}_i^t - \bar{\boldsymbol{\mu}}_i^* \rangle \leq \frac{\sum_{k=1}^{K} \mathbb{E}\|\boldsymbol{\mu}_k^t - \boldsymbol{\mu}_k^*\|_{\boldsymbol{\Sigma}_k^{t-1}}^2 - \sum_{k=1}^{K} \mathbb{E}\|\boldsymbol{\mu}_k^{t+1} - \boldsymbol{\mu}_k^*\|_{\boldsymbol{\Sigma}_k^{t-1}}^2}{2\beta_t} + \beta_t \sum_{k=1}^{K} \mathbb{E}\|\boldsymbol{\Sigma}_k^t (\sum_{i=k}^{K} \hat{g}_{ik})\|_{\boldsymbol{\Sigma}_k^{t-1}}^2$$

$$- \sum_{k=1}^{K} \frac{\gamma_t}{2} (\mathbb{E}\|\boldsymbol{\mu}_k^t - \boldsymbol{\mu}_k^*\|_2^2 - \|\boldsymbol{\mu}_k^*\|_2^2) \tag{126}$$

From Lemma C.4, we know that for $\forall i \in \{1, \cdots, K\}$, $J_i(\bar{\boldsymbol{\mu}}_i, \bar{\boldsymbol{\Sigma}}_i)$ is convex function w.r.t. $\bar{\boldsymbol{\mu}}_i$ and $\bar{\boldsymbol{\Sigma}}_i^{\frac{1}{2}}$. Then, we have that

$$J_i(\bar{\boldsymbol{\mu}}_i^t, \bar{\boldsymbol{\Sigma}}_i^t) - J_i(\bar{\boldsymbol{\mu}}_i^*, 0) \leq \langle g_i^t, \bar{\boldsymbol{\mu}}_i^t - \bar{\boldsymbol{\mu}}_i^* \rangle + \langle \nabla_{\bar{\boldsymbol{\Sigma}}_i^{\frac{1}{2}} = \bar{\boldsymbol{\Sigma}}_i^{t\frac{1}{2}}} J_i(\bar{\boldsymbol{\mu}}_i^t, \bar{\boldsymbol{\Sigma}}_i^t), \bar{\boldsymbol{\Sigma}}_i^{t\frac{1}{2}} - 0 \rangle \tag{127}$$

Denote $G_i^t = \nabla_{\bar{\boldsymbol{\Sigma}}_i = \bar{\boldsymbol{\Sigma}}_i^t} J_i(\bar{\boldsymbol{\mu}}_i^t, \bar{\boldsymbol{\Sigma}}_i^t)$. Note that $\nabla_{\bar{\boldsymbol{\Sigma}}_i^{t\frac{1}{2}}} J_i = \bar{\boldsymbol{\Sigma}}_i^{t\frac{1}{2}} \nabla_{\bar{\boldsymbol{\Sigma}}_i^t} J_i + \nabla_{\bar{\boldsymbol{\Sigma}}_i^t} J_i \bar{\boldsymbol{\Sigma}}_i^{t\frac{1}{2}}$, and $G_i^t$, $\nabla_{\bar{\boldsymbol{\Sigma}}_i^t} J_i$ and $\bar{\boldsymbol{\Sigma}}_i^{t\frac{1}{2}}$ are symmetric matrix, it follows that

$$\langle \nabla_{\bar{\boldsymbol{\Sigma}}_i^{\frac{1}{2}} = \bar{\boldsymbol{\Sigma}}_i^{t\frac{1}{2}}} J_i(\bar{\boldsymbol{\mu}}_i^t, \bar{\boldsymbol{\Sigma}}_i^t), \bar{\boldsymbol{\Sigma}}_i^{t\frac{1}{2}} - 0 \rangle = \langle \bar{\boldsymbol{\Sigma}}_i^{t\frac{1}{2}} G_i^t + G_i^t \bar{\boldsymbol{\Sigma}}_i^{t\frac{1}{2}}, \bar{\boldsymbol{\Sigma}}_i^{t\frac{1}{2}} \rangle \tag{128}$$

$$= 2 \langle G_i^t, \bar{\boldsymbol{\Sigma}}_i^t \rangle = 2\text{tr}(G_i^t \bar{\boldsymbol{\Sigma}}_i^t) \tag{129}$$

Plug Eq.(129) into Eq. (127), we have that

$$J_i(\bar{\boldsymbol{\mu}}_i^t, \bar{\boldsymbol{\Sigma}}_i^t) - J_i(\bar{\boldsymbol{\mu}}_i^*, 0) \leq \langle g_i^t, \bar{\boldsymbol{\mu}}_i^t - \bar{\boldsymbol{\mu}}_i^* \rangle + 2\text{tr}(G_i^t \bar{\boldsymbol{\Sigma}}_i^t) \tag{130}$$

It follows that

$$\sum_{i=1}^{K} J_i(\bar{\boldsymbol{\mu}}_i^t, \bar{\boldsymbol{\Sigma}}_i^t) - \sum_{i=1}^{K} J_i(\bar{\boldsymbol{\mu}}_i^*, 0) \leq \sum_{i=1}^{K} \langle g_i^t, \bar{\boldsymbol{\mu}}_i^t - \bar{\boldsymbol{\mu}}_i^* \rangle + 2 \sum_{i=1}^{K} \text{tr}(G_i^t \bar{\boldsymbol{\Sigma}}_i^t) \tag{131}$$

Plug Eq.(126) into Eq.(131), we have that

$$\sum_{i=1}^{K} J_i(\bar{\boldsymbol{\mu}}_i^t, \bar{\boldsymbol{\Sigma}}_i^t) - \sum_{i=1}^{K} J_i(\bar{\boldsymbol{\mu}}_i^*, 0)$$

$$\leq \frac{\sum_{k=1}^{K} \mathbb{E}\|\boldsymbol{\mu}_k^t - \boldsymbol{\mu}_k^*\|_{\boldsymbol{\Sigma}_k^{t-1}}^2 - \sum_{k=1}^{K} \mathbb{E}\|\boldsymbol{\mu}_k^{t+1} - \boldsymbol{\mu}_k^*\|_{\boldsymbol{\Sigma}_k^{t-1}}^2 - \beta_t \gamma_t \sum_{i=1}^{K} \mathbb{E}\|\boldsymbol{\mu}_i^t - \boldsymbol{\mu}_i^*\|_2^2}{2\beta_t}$$

$$+ \beta_t \sum_{k=1}^{K} \mathbb{E}\|\boldsymbol{\Sigma}_k^t (\sum_{i=k}^{K} \hat{g}_{ik})\|_{\boldsymbol{\Sigma}_k^{t-1}}^2 + 2 \sum_{i=1}^{K} \text{tr}(G_i^t \bar{\boldsymbol{\Sigma}}_i^t) + \frac{\gamma_t}{2} \sum_{k=1}^{K} \|\boldsymbol{\mu}_k^*\|_2^2 \tag{132}$$

In addition, we have that

$$\frac{1}{\beta_{t+1}} \mathbb{E}\|\boldsymbol{\mu}_k^{t+1} - \boldsymbol{\mu}_k^*\|_{\boldsymbol{\Sigma}_k^{t+1-1}}^2 - \frac{1}{\beta_t} \mathbb{E}\|\boldsymbol{\mu}_k^{t+1} - \boldsymbol{\mu}_k^*\|_{\boldsymbol{\Sigma}_k^{t-1}}^2 - \gamma_t \mathbb{E}\|\boldsymbol{\mu}_i^t - \boldsymbol{\mu}_i^*\|_2^2$$

$$= \frac{1}{\beta_{t+1}} \mathbb{E}\langle \boldsymbol{\Sigma}_k^{t+1-1}(\boldsymbol{\mu}_k^{t+1} - \boldsymbol{\mu}_k^*), \boldsymbol{\mu}_k^{t+1} - \boldsymbol{\mu}_k^* \rangle - \frac{1}{\beta_t} \mathbb{E}\langle \boldsymbol{\Sigma}_k^{t-1}(\boldsymbol{\mu}_k^{t+1} - \boldsymbol{\mu}_k^*), \boldsymbol{\mu}_k^{t+1} - \boldsymbol{\mu}_k^* \rangle - \gamma_t \mathbb{E}\|\boldsymbol{\mu}_i^t - \boldsymbol{\mu}_i^*\|_2^2 \tag{133}$$

$$= \mathbb{E}\langle \left(\frac{1}{\beta_{t+1}} \boldsymbol{\Sigma}_k^{t+1-1} - \frac{1}{\beta_t} \boldsymbol{\Sigma}_k^{t-1} - \gamma_t \boldsymbol{I}\right)(\boldsymbol{\mu}_k^{t+1} - \boldsymbol{\mu}_k^*), \boldsymbol{\mu}_k^{t+1} - \boldsymbol{\mu}_k^* \rangle \tag{134}$$

Note that $\boldsymbol{\Sigma}_k^{t+1-1} = \omega_t \boldsymbol{\Sigma}_k^{t-1} + \alpha_t \hat{G}_k^t = \boldsymbol{\Sigma}_k^{t-\frac{1}{2}}(\omega_t \boldsymbol{I} + \alpha_t \boldsymbol{H}_k^t)\boldsymbol{\Sigma}_k^{t-\frac{1}{2}}$, we have that

$$\frac{1}{\beta_{t+1}} \boldsymbol{\Sigma}_k^{t+1-1} - \frac{1}{\beta_t} \boldsymbol{\Sigma}_k^{t-1} - \gamma_t \boldsymbol{I} = \frac{1}{\beta_{t+1}} \boldsymbol{\Sigma}_k^{t-\frac{1}{2}}(\omega_t \boldsymbol{I} + \alpha_t \boldsymbol{H}_k^t)\boldsymbol{\Sigma}_k^{t-\frac{1}{2}} - \frac{1}{\beta_t} \boldsymbol{\Sigma}_k^{t-1} - \gamma_t \boldsymbol{I} \tag{135}$$

$$= \boldsymbol{\Sigma}_k^{t-\frac{1}{2}}\left(\frac{\omega_t}{\beta_{t+1}} \boldsymbol{I} + \frac{\alpha_t}{\beta_{t+1}} \boldsymbol{H}_k^t - \frac{1}{\beta_t} \boldsymbol{I} - \gamma_t \boldsymbol{\Sigma}_k^t\right)\boldsymbol{\Sigma}_k^{t-\frac{1}{2}} \tag{136}$$

Because of $\boldsymbol{H}_k^t \preceq \frac{1}{\alpha_t}(\frac{\beta_{t+1}}{\beta_t} - \omega_t)\boldsymbol{I} + \frac{\beta_{t+1}\gamma_t}{\alpha_t}\boldsymbol{\Sigma}_k^t$ in algorithm 2, we have that

$$\frac{1}{\beta_{t+1}}\boldsymbol{\Sigma}_k^{t+1^{-1}} - \frac{1}{\beta_t}\boldsymbol{\Sigma}_k^{t^{-1}} - \gamma_t\boldsymbol{I} = \boldsymbol{\Sigma}_k^{t^{-\frac{1}{2}}}(\frac{\omega_t}{\beta_{t+1}}\boldsymbol{I} + \frac{\alpha_t}{\beta_{t+1}}\boldsymbol{H}_k^t - \frac{1}{\beta_t}\boldsymbol{I} - \gamma_t\boldsymbol{\Sigma}_k^t)\boldsymbol{\Sigma}_k^{t^{-\frac{1}{2}}} \tag{137}$$

$$\preceq \boldsymbol{\Sigma}_k^{t^{-\frac{1}{2}}}(\frac{\omega_t}{\beta_{t+1}}\boldsymbol{I} + \frac{1}{\beta_{t+1}}(\frac{\beta_{t+1}}{\beta_t} - \omega_t)\boldsymbol{I} + \gamma_t\boldsymbol{\Sigma}_k^t - \frac{1}{\beta_t}\boldsymbol{I} - \gamma_t\boldsymbol{\Sigma}_k^t)\boldsymbol{\Sigma}_k^{t^{-\frac{1}{2}}} \tag{138}$$

$$\preceq \boldsymbol{0} \tag{139}$$

Plug Eq.(139) into Eq.(134), we know that

$$\frac{1}{\beta_{t+1}}\mathbb{E}\|\boldsymbol{\mu}_k^{t+1} - \boldsymbol{\mu}_k^*\|_{\boldsymbol{\Sigma}_k^{t+1-1}}^2 - \frac{1}{\beta_t}\mathbb{E}\|\boldsymbol{\mu}_k^{t+1} - \boldsymbol{\mu}_k^*\|_{\boldsymbol{\Sigma}_k^{t-1}}^2 - \gamma_t\mathbb{E}\|\boldsymbol{\mu}_i^t - \boldsymbol{\mu}_i^*\|_2^2 \leq 0 \tag{140}$$

Telescope with Eq.(132), we have that

$$\sum_{t=1}^T \sum_{i=1}^K J_i(\bar{\boldsymbol{\mu}}_i^t, \bar{\boldsymbol{\Sigma}}_i^t) - \sum_{t=1}^T \sum_{i=1}^K J_i(\bar{\boldsymbol{\mu}}_i^*, 0)$$
$$\leq \frac{\sum_{k=1}^K \mathbb{E}\|\boldsymbol{\mu}_k^1 - \boldsymbol{\mu}_k^*\|_{\boldsymbol{\Sigma}_k^{1-1}}^2}{2\beta_1} - \frac{\sum_{k=1}^K \mathbb{E}\|\boldsymbol{\mu}_k^{T+1} - \boldsymbol{\mu}_k^*\|_{\boldsymbol{\Sigma}_k^{T-1}}^2}{2\beta_T} + \sum_{t=1}^T \beta_t \sum_{k=1}^K \mathbb{E}\|\boldsymbol{\Sigma}_k^t(\sum_{i=k}^K \hat{g}_{ik})\|_{\boldsymbol{\Sigma}_k^{t-1}}^2$$
$$+ 2\sum_{t=1}^T \sum_{i=1}^K \text{tr}(G_i^t\bar{\boldsymbol{\Sigma}}_i^t) + \frac{1}{2}\sum_{t=1}^T \gamma_t \sum_{k=1}^K \|\boldsymbol{\mu}_k^*\|_2^2 \tag{141}$$

We now show the upper bound of term $\sum_{t=1}^T \beta_t \sum_{k=1}^K \mathbb{E}\|\boldsymbol{\Sigma}_k^t(\sum_{i=k}^K \hat{g}_{ik})\|_{\boldsymbol{\Sigma}_k^{t-1}}^2$.

Note that $\beta_t = t\beta$, together with Lemma C.2 (c), we know that

$$\sum_{t=1}^T \beta_t \sum_{k=1}^K \mathbb{E}\|\boldsymbol{\Sigma}_k^t(\sum_{i=k}^K \hat{g}_{ik})\|_{\boldsymbol{\Sigma}_k^{t-1}}^2 \leq \sum_{t=1}^T t\frac{C_1}{t^{\frac{3}{2}}} \leq 2\sqrt{T+1}C_1 \tag{142}$$

where $C_1 = \frac{3\beta\sum_{i=1}^K KL_i^2(id+1)^2}{2\alpha\nu}$.

We now show the upper bound of term $2\sum_{t=1}^T \sum_{i=1}^K \text{tr}(G_i^t\bar{\boldsymbol{\Sigma}}_i^t)$.

From Lemma C.3, we know that

$$2\sum_{t=1}^T \sum_{i=1}^K \text{tr}(G_i^t\bar{\boldsymbol{\Sigma}}_i^t) \leq 2\sum_{t=1}^T \sum_{i=1}^K \frac{L_i id}{2t^{\frac{3}{4}}}\sqrt{\frac{3}{\alpha\nu}} \tag{143}$$

$$= \sum_{t=1}^T C_2\frac{1}{t^{\frac{3}{4}}} \tag{144}$$

$$\leq 4(T+1)^{\frac{1}{4}}C_2 \tag{145}$$

where $C_2 = \frac{\sum_{i=1}^K \sqrt{3}idL_i}{\sqrt{\alpha\nu}}$ .

We now show the upper bound of the term $\frac{1}{2}\sum_{t=1}^T \gamma_t \sum_{k=1}^K \|\boldsymbol{\mu}_k^*\|_2^2$.

Note that the optimal solution is bounded from Assumption 5.3, i.e., $\sum_{k=1}^K \|\boldsymbol{\mu}_k^*\|_2^2 \leq B$. Together with the setting $\gamma_t = \frac{\alpha\nu}{\beta\sqrt{t+1}}$, we can achieve that

$$\frac{1}{2}\sum_{t=1}^T \gamma_t \sum_{k=1}^K \|\boldsymbol{\mu}_k^*\|_2^2 \leq \frac{B}{2}\sum_{t=1}^T \gamma_t \leq \frac{\alpha\nu B}{2\beta}2\sqrt{T+2} = \sqrt{T+2}C_3 \tag{146}$$

where $C_3 = \frac{\alpha\nu B}{\beta}$

Plug Eq.(142) and Eq.(145) into Eq.(141), we have that

$$
\sum_{t=1}^{T}\sum_{i=1}^{K} J_i(\bar{\boldsymbol{\mu}}_i^t, \bar{\boldsymbol{\Sigma}}_i^t) - \sum_{t=1}^{T}\sum_{i=1}^{K} J_i(\bar{\boldsymbol{\mu}}_i^*, 0)
$$
$$
\leq \frac{\sum_{k=1}^{K} \mathbb{E}\|\boldsymbol{\mu}_k^1 - \boldsymbol{\mu}_k^*\|^2_{\boldsymbol{\Sigma}_k^{1-1}}}{2\beta_1} - \frac{\sum_{k=1}^{K} \mathbb{E}\|\boldsymbol{\mu}_k^{T+1} - \boldsymbol{\mu}_k^*\|^2_{\boldsymbol{\Sigma}_k^{T-1}}}{2\beta_T} + 2\sqrt{T+1}C_1 + 4(T+1)^{\frac{1}{4}}C_2 + \sqrt{T+2}C_3
$$
$$(147)$$

From Lemma C.5, we then have that

$$
\sum_{t=1}^{T}\left(\sum_{k=1}^{K}(f_k(\bar{\boldsymbol{\mu}}_k^t) - f_k(\bar{\boldsymbol{\mu}}_k^*))\right) \leq \sum_{t=1}^{T}\sum_{i=1}^{K} J_i(\bar{\boldsymbol{\mu}}_i^t, \bar{\boldsymbol{\Sigma}}_i^t) - \sum_{t=1}^{T}\sum_{i=1}^{K} J_i(\bar{\boldsymbol{\mu}}_i^*, 0) \tag{148}
$$
$$
\leq \frac{\sum_{k=1}^{K} \mathbb{E}\|\boldsymbol{\mu}_k^1 - \boldsymbol{\mu}_k^*\|^2_{\boldsymbol{\Sigma}_k^{1-1}}}{2\beta_1} - \frac{\sum_{k=1}^{K} \mathbb{E}\|\boldsymbol{\mu}_k^{T+1} - \boldsymbol{\mu}_k^*\|^2_{\boldsymbol{\Sigma}_k^{T-1}}}{2\beta_T}
$$
$$
+ +2\sqrt{T+1}C_1 + 4(T+1)^{\frac{1}{4}}C_2 + \sqrt{T+2}C_3 \tag{149}
$$

Finally, divide $T$ on both sides of Eq.(149), we have that

$$
\frac{1}{T}\sum_{t=1}^{T}\sum_{k=1}^{K} f_k(\bar{\boldsymbol{\mu}}_k^t) - \sum_{k=1}^{K} f_k(\bar{\boldsymbol{\mu}}_k^*) \leq \frac{\sum_{k=1}^{K}\|\boldsymbol{\mu}_k^1 - \boldsymbol{\mu}_k^*\|^2_{\boldsymbol{\Sigma}_k^{1-1}}}{2\beta T} + \frac{2\sqrt{T+1}C_1}{T} + \frac{4(T+1)^{\frac{1}{4}}C_2}{T} + \frac{\sqrt{T+2}C_3}{T}
$$
$$(150)$$

$$\square$$

**Theorem D.2.** *Set $\beta_t = t\beta$ with $\beta > 0$, $\alpha_t = \sqrt{t+1}\alpha$ with $\alpha > 0$, and $\gamma_t = \frac{\alpha\nu}{\beta\sqrt{t+1}}$, and $\nu > 0$, and $\omega_t = 1$. Then, during the running process of Algorithm 2, the constraints $\boldsymbol{H}_k^t \preceq \frac{1}{\alpha_t}(\frac{\beta_{t+1}}{\beta_t} - \omega_t)\boldsymbol{I} + \frac{\beta_{t+1}\gamma_t}{\alpha_t}\boldsymbol{\Sigma}_k^t$ and $\nu\boldsymbol{I} \preceq \hat{G}_k^t = \boldsymbol{\Sigma}_k^{t-\frac{1}{2}}\boldsymbol{H}_k^t\boldsymbol{\Sigma}_k^{t-\frac{1}{2}}$ always have feasible solutions.*

*Proof.* To ensure the constraints $\boldsymbol{H}_k^t \preceq \frac{1}{\alpha_t}(\frac{\beta_{t+1}}{\beta_t} - \omega_t)\boldsymbol{I} + \frac{\beta_{t+1}\gamma_t}{\alpha_t}\boldsymbol{\Sigma}_k^t$ and $\nu\boldsymbol{I} \preceq \hat{G}_k^t = \boldsymbol{\Sigma}_k^{t-\frac{1}{2}}\boldsymbol{H}_k^t\boldsymbol{\Sigma}_k^{t-\frac{1}{2}}$ always feasible during the algorithm, it is equivalent to show the constraints Eq.(151) always has feasible solutions $\boldsymbol{H}_k^t$.

$$
\nu\boldsymbol{\Sigma}_k^t \preceq \boldsymbol{H}_k^t \preceq \frac{1}{\alpha_t}(\frac{\beta_{t+1}}{\beta_t} - \omega_t)\boldsymbol{I} + \frac{\beta_{t+1}\gamma_t}{\alpha_t}\boldsymbol{\Sigma}_k^t \tag{151}
$$

It is equivalent to show that the inequality (152) always holds true.

$$
\nu\boldsymbol{\Sigma}_k^t \preceq \frac{1}{\alpha_t}(\frac{\beta_{t+1}}{\beta_t} - \omega_t)\boldsymbol{I} + \frac{\beta_{t+1}\gamma_t}{\alpha_t}\boldsymbol{\Sigma}_k^t \tag{152}
$$

Then, it is equivalent to show that the inequality (153) always holds true.

$$
(\nu - \frac{\beta_{t+1}\gamma_t}{\alpha_t})\boldsymbol{\Sigma}_k^t \preceq \frac{1}{\alpha_t}(\frac{\beta_{t+1}}{\beta_t} - \omega_t)\boldsymbol{I} \tag{153}
$$

We first check the left hand side of the inequality (153).

Note that the setting $\beta_t = t\beta$ with $\beta > 0$, $\alpha_t = \sqrt{t+1}\alpha$ with $\alpha > 0$, and $\gamma_t = \frac{\alpha\nu}{\beta\sqrt{t+1}}$, and $\nu > 0$, and $\omega_t = 1$. We can achieve that

$$
(\nu - \frac{\beta_{t+1}\gamma_t}{\alpha_t})\boldsymbol{\Sigma}_k^t = (\nu - \frac{(t+1)\beta}{\sqrt{t+1}\alpha}\frac{\alpha\nu}{\beta\sqrt{t+1}})\boldsymbol{\Sigma}_k^t \tag{154}
$$
$$
= (\nu - \nu)\boldsymbol{\Sigma}_k^t = 0 \tag{155}
$$

We now check the right hand side of the inequality (153).

$$\frac{1}{\alpha_t}(\frac{\beta_{t+1}}{\beta_t} - \omega_t) = \frac{1}{\alpha_t}(\frac{(t+1)\beta}{t\beta} - 1) = \frac{1}{\alpha_t t} > 0 \tag{156}$$

Thus, the inequality (153) always hold true. As a result, the constraints set always have feasible solutions.

$\square$

## E  DETAILS OF DIFFUSION TARGET GENERATION EXPERIMENTS

### E.1  BLACK-BOX TARGET FUNCTION

CLIP (Contrastive Language-Image Pre-Training) (Radford et al., 2021) is trained on various (image, text) pairs to capture their similarity. Let the generated image be $x$, the target text be $y$, and the image and text encoders of CLIP be $\phi_{\text{image}}$ and $\phi_{\text{text}}$, respectively. These encoders share an aligned embedding space, allowing us to measure similarity using cosine distance. The black-box function is defined as:

$$f(x; y) = \frac{\phi_{image}(x)^T \phi_{text}(y)}{\|\phi_{image}(x)\|\|\phi_{text}(y)\|}$$

We then employ the normalized cosine distance $\frac{y-\mu_y}{\sigma_y}$ as the black-box target score, where $\mu_y$ and $\sigma_y$ denotes the mean and standard deviation of the cosine distances between the target and the images in the dataset.

For our implementation, we choose to use the pre-trained publicly available CLIP model ViT-L/14[3].

### E.2  SAMPLER: DPM-SOLVER++

We choose to use DPM-Solver++ (Lu et al., 2022a) as our sampler for all experiments in both the training and evaluation phases. We use the 2nd-order SDE solver and the data evaluation formulation. For all experiments, we use $K = 14$ sampling steps.

### E.3  DATASET: CELEBA-HQ

We use dataset CelebA-HQ in our experiments, it contains 30,000 face images. It is public avaliable at https://huggingface.co/datasets/huggan/CelebA-HQ

### E.4  BASELINE: CLASSIFIER-FREE CONDITIONAL DIFFUSION MODEL

Denotes the dataset as $\mathcal{X} = \{x_1, x_2, ..., x_n\}$. We evaluate the black-box function target score value, that is $\mathcal{Y} = \{f(x_i) \mid \forall x_i \in \mathcal{X}\}$, and denotes the mean and standard deviation as $\mu_y$ and $\sigma_y$. We then obtain the normalized target score value as $\widetilde{\mathcal{Y}} = \{\frac{y_i - \mu_y}{\sigma_y} \mid \forall y_i \in \mathcal{Y}\}$.

We use the dataset and normalized score value $\mathcal{X} \times \widetilde{\mathcal{Y}}$ to train a classifier-free conditional diffusion model $\hat{\mu}_\phi(x, t, y)$.

An unconditional diffusion model trained on the CelebA-HQ dataset is provided by the authors of latent diffusion [4]. In practice, instead of training the model from scratch, we load the weights of the unconditional layers from the pre-trained model and continue training from there. This approach can speed up the training time.

---

[3]https://huggingface.co/sentence-transformers/clip-ViT-L-14
[4]https://ommer-lab.com/files/latent-diffusion/celeba.zip

## E.5   BASELINE: DDOM

The recent work (Krishnamoorthy et al., 2023) introduces Denoising Diffusion Optimization Models (DDOM) for solving offline black-box optimization tasks using diffusion models. This method can also be naturally extended to black-box targeted generation tasks.

During the pre-processing phase, the black-box function score $y_i = f(x_i)$ is evaluated for each sample $x_i \in \mathcal{D}$. The offline dataset $\mathcal{D}$ is then partitioned into $N_B$ bins of equal width based on $y$. Each bin is assigned a weight proportional to both the number of points in the bin and the average score value of the bin. Specifically, the weight of the $i$-th is given by:

$$w_i = \frac{|B_i|}{|B_i| + C} \exp\left(\frac{-|\hat{y} - y_{b_i}|}{\tau}\right) \tag{157}$$

where $\hat{y}$ is the best function value in the offline dataset $\mathcal{D}$, $|B_i|$ denotes the number of points in the $i$-th bin, and $y_{b_i}$ is the midpoint of the interval corresponding to the bin $B_i$. The parameters $K$ and $\tau$ are hyper-parameters.

During training, this weight is used to compute the weighted loss, which is given by:

$$\mathbb{E}_t\left[\lambda(t) \mathop{\mathbb{E}}_{\mathbf{x}_0, y}\left[w(y) \mathop{\mathbb{E}}_{\mathbf{x}_t|\mathbf{x}_0}\left[\|\epsilon_\theta(\mathbf{x}_t, t, y) - \nabla_{\mathbf{x}} \log p_t(\mathbf{x}_t \mid \mathbf{x}_0)\|_2^2\right]\right]\right] \tag{158}$$

where $w(y) = w_i$ if $y \in B_i$. In practice, the score values are normalized to fit a standard normal distribution to ensure that the $y_i$ values are well behaved.

The authors demonstrate DDOM for solving black-box optimization (BBO) tasks. Since DDOM uses a diffusion model, it can also be naturally implemented for image generation tasks. In our experiment, we implement the DDOM for black-box targeted image generation tasks. In the weight function defined in equation (157), we set the hyper-parameters $C = 0.01$, $\tau = 0.1$, $N_B = 64$.

## E.6   OUR METHOD: FINE-TUNE THE CLASSIFIER-FREE CONDITIONAL DIFFUSION MODEL

We use our method to improve the baseline conditional diffusion model. Instead of fine-tune the parameters for all time steps, we only apply it for the second half of the time steps. The rationale is that we should place more emphasis on the time steps closer to the final output image $x_K$. That is, we have the fine-tuning parameter sets $\theta_k = \{\mu_k, \Sigma_k\}$ for $k \in \{1, ..., K\}$.

We revise the sequential black-box objective $\sum_{k=1}^K f_k(\bar{\epsilon}_k)$. In our previous experiments, we set $f_k(\bar{\epsilon}_k) = F(\boldsymbol{x}_k)$ for $k \in \{1, \cdots, K\}$, which call CLIP to evaluate the input (noised) image $\boldsymbol{x}_k$ at each diffusion sampling step $k$. This scheme is not effective enough because we care more about the generated image at the last step, i.e., $\boldsymbol{x}_K$, than the generated image at the inner steps. Thus, we set the black-box function at the inner step as $f_k(\bar{\epsilon}_k) = F_k(\boldsymbol{x}_k)$ for $k \in \{1, \cdots, K-1\}$. The black-box function $F_k(\boldsymbol{x}_k)$ takes the noised image $\boldsymbol{x}_k$ at the inner step as input and performs a deterministic sampling process $\boldsymbol{x}_{k+1} = \hat{\boldsymbol{\mu}}_\phi(\boldsymbol{x}_k, k) + \tilde{\sigma}_{k+1}\boldsymbol{\mu}_{k+1}$ for the future steps to achieve a predicted $\boldsymbol{x}_K$, and call CLIP to evaluate the predicted $\boldsymbol{x}_K$.

