# OpenReview forum: "Stochastic Adaptive Sequential  Black-box Optimization for  Diffusion Targeted Generation"
_ICLR.cc/2024/Conference — Submitted to ICLR 2024_

### Official Review · Reviewer_djVx · 2023-10-31

**Soundness:** 3 good
**Presentation:** 2 fair
**Contribution:** 2 fair
**Rating:** 5
**Confidence:** 2

**Summary:**

The paper discusses a significant challenge in performing user-preferred targeted generation via diffusion models with only black-box target scores of users. They address this challenge by formulating the fine-tuning of the inference phase of a pre-trained diffusion model
as a sequential black-box optimization problem. In practice, they propose a stochastic adaptive sequential optimization algorithm to optimize cumulative black-box scores under unknown transition dynamics. Theoretical and empirical evidence are provided to support the method.

**Strengths:**

* The formulation regarding the fine-tuning of the inference of diffusion models as a black-box sequential optimization problem is sound and novel, which may inspire subsequent works in this area.
* The paper includes an extensive theoretical analysis to validate the convergence of the proposed method and its superior ability to manage non-smooth problems.

**Weaknesses:**

* The methodology, as currently presented, might be heavy and unfriendly to the general audience. The authors should aim to articulate their primary contributions and the uniqueness of their proposed methodology more clearly, focusing on how it addresses the identified challenges.
* The empirical results section could be enhanced.
     * The rationale for using text-guided generation as a demonstration of the proposed method is not entirely clear. The authors should elaborate on why they believe this setting poses a challenging black-box optimization score. Does this scorer belong to those touch non-smooth cases?
    * The quality of generated images is damaged. As evidenced in Figure 2, the generated images of baseline methods are more natural and clear than those of the proposed method. Is there any solution to resolve this problem?
    * The single-domain experiment on text-guided generation does not sufficiently prove the effectiveness of the method proposed. The authors should consider introducing more diverse black-box scorers to exhibit the method's versatility. For example, DDOM conducted experiments across multiple domains; perhaps a similar approach could be beneficial here.

**Questions:**

* It is recommended that the authors revise the methodology section to accentuate the main technical challenge they've addressed and the motivation behind their method.
* The derivation process in Equation (10) and how this objective aligns with Equation (8) is not clear. Can the authors provide further clarification on this?
* Certain notations, such as $\alpha$ and $s$ in Section 2.1, lack explanation and should be clarified for reader comprehension.

---

> ### Author Response · Authors · 2023-11-22
>
> We sincerely appreciate the reviewer's constructive advice and valuable comments. Our detailed responses to the reviewer's questions are presented as follows:
>
> $\textbf{Q1}:$ "The authors should aim to articulate their primary contributions and the uniqueness of their proposed methodology more clearly, focusing on how it addresses the identified challenges."
>
>
> $\textbf{A1:}$ In this paper, we focus on the problem of how to employ diffusion models to generate user-preferred content with black-box target scores while avoiding re-training. This problem is essential but unexplored in the literature. The related DDOM needs a reweighted sampling of training samples to re-train a conditional diffusion model from scratch.
>
>
> Moreover, the black-box optimization methods in the literature can not be directly applied for diffusion model fine-tuning due to ignoring the sequential nature of the diffusion sampling process. To address this problem, we $ \textbf{are the first to}$ formulate the diffusion model black-box fine-tuning problem as a sequential black-box optimization problem with an adaptive diffusion noise process.
> We propose a novel covariance adaptive sequential black-box optimization algorithm to solve this problem. Furthermore, we prove a $O(\frac{d^2}{\sqrt{T}})$ convergence rate of our algorithm for convex functions without smooth and strongly convex assumptions. To the best of our knowledge, our algorithm is the $\textbf{first}$  covariance adaptive black-box optimization method that achieves a provable $O(\frac{d^2}{\sqrt{T}})$ convergence rate for general non-smooth convex functions.
>
>
> We firmly believe that our work explores a crucial area underexplored and provides solid technique contributions to this area.
>
>
>
>
> $\textbf{Q2:}$ "The rationale for using text-guided generation as a demonstration of the proposed method is not entirely clear. The authors should elaborate on why they believe this setting poses a challenging black-box optimization score. Does this scorer belong to those touch non-smooth cases?"
>
> $\textbf{A2:}$ The black-box targeted image generation is challenging due to the very high dimension of images.  Specifically,   the dimension of the images employed in our work is $256\times256\times3 = 196608$.  In contrast, most of the tasks employed in the DDOM paper are less than $100$-dimension, and the highest dimension is $5152$.  The image generation task is more challenging than the one studied in the DDOM paper.
>
> "Does this scorer belong to those touch non-smooth cases?"
>
> Most scorers in real applications are non-smooth cases. For example, the scorers based on neural networks with ReLU function are non-smooth. To be more precise,  note that the ReLU function is non-smooth. Thus, the neural network using the ReLU function is also non-smooth. As a result,  the score functions that rely on  the neural network  using the ReLU function are also non-smooth.
>
>
> $\textbf{Q3:}$ "The quality of generated images is damaged. As evidenced in Figure 2, the generated images of baseline methods are more natural and clear than those of the proposed method. Is there any solution to resolve this problem?"
>
> $\textbf{A3:}$ We have solved the loss of visual quality issue and include additional experiments.  Please check our overall response and Section A in the Appendix of our revised paper.
>
>
> $\textbf{Q4:}$ "The single-domain experiment on text-guided generation does not sufficiently prove the effectiveness of the method proposed. The authors should consider introducing more diverse black-box scorers to exhibit the method's versatility. For example, DDOM conducted experiments across multiple domains; perhaps a similar approach could be beneficial here.
> "
>
> $\textbf{A4:}$ Thanks for your suggestion. We include additional experiments on the tasks studied in the DDOM. Please check our overall response and Section A in the Appendix of our revised paper.
>
>
>
> $\textbf{Q5:}$ "The derivation process in Equation (10) and how this objective aligns with Equation (8) is not clear. Can the authors provide further clarification on this?"
>
>
> $\textbf{A5:}$ Note that $J(\bar{\theta}_K^t)$ is a constant when we minimize $J(\bar{\theta}_K)$ w.r.t.  $\bar{\theta}_K$. Thus, minimizing Eq.(10) (Eq.(9) in the revised paper)  is equivalent to minimizing objective Eq.(8) (Eq.(7) in the revised paper)

---

### Official Review · Reviewer_NXdQ · 2023-10-31

**Soundness:** 3 good
**Presentation:** 1 poor
**Contribution:** 3 good
**Rating:** 6
**Confidence:** 2

**Summary:**

This paper proposed a novel algorithm (stochastic adaptive black box sequential optimization) to generate user targeted samples.   The authors provided a theoretical analysis on the convergence rate as well as other properties.  Empirically, the experiment results on CelebA-HD datasets showed a clear improvement on generating more targeted images.  However, this comes at a cost of perception quality.

**Strengths:**

The paper has a solid theoretical ground, and proposed a novel approach.  The authors are able to validate the theory by their empirical results.

**Weaknesses:**

1.  The organisation and writing of the paper need to be improved.  The notations and equations need to be better defined.  For example, the authors should define all terms appeared in equation 1, 2, and 3, such as g(t) and \alpha_t and so on.  The writing made it very difficult for readers to follow the proposed method.

2.  The loss of perception quality of the targeted generation is a hurdle for this approach to have any practical use.  It'd be better if the authors explored this trade-off a bit further.

**Questions:**

What do you think has caused the loss of perception quality in the targeted generation? You mentioned the solution could be adding a quality measurement (I'm guessing something like LPIPS)  into the black-box function.  Do you have any insight in what will happen? Would it help the perception quality while keeping the target score high? Or would it be a tradeoff?

---

> ### Author Response · Authors · 2023-11-22
>
> We sincerely appreciate the reviewer's constructive advice and valuable comments. Our detailed responses to the reviewer's questions are presented as follows:
>
> $\textbf{Q1:}$ "The organisation and writing of the paper need to be improved. The notations and equations need to be better defined. For example, the authors should define all terms appeared in equation 1, 2, and 3, such as $g(t)$ and $\alpha_t$ and so on. The writing made it very difficult for readers to follow the proposed method."
>
> $\textbf{A1:}$ Thanks for your suggestion. We have revised the paper accordingly. The notation in Eq.(1)~(2) is clearly defined in the revised paper.
>
> $\textbf{Q2:}$ "The loss of perception quality of the targeted generation is a hurdle for this approach to have any practical use. It'd be better if the authors explored this trade-off a bit further."
>
>
> $\textbf{A2:}$ We solve the loss of visual quality issue and include additional experiments.  Please check our overall response and Section A in the Appendix of our revised paper.

---

### Official Review · Reviewer_q9dy · 2023-10-31

**Soundness:** 2 fair
**Presentation:** 1 poor
**Contribution:** 2 fair
**Rating:** 3
**Confidence:** 3

**Summary:**

This paper presents method to fine-tune the inference phase of a pre-trained diffusion model as a sequential black-box optimization problem in order to maximize some pre-defined score function. The paper proves a $O(d^2/\sqrt{T})$ convergence rate for cumulative convex functions. Empirically, the proposed method is evaluated on a text-to-image generation tasks and yields higher target scores compared to other approaches.

**Strengths:**

The biggest strength of the paper lies in its technical novelty and motivation to solve the targeted generation problems with diffusion models using an adaptive sequential scheme. By utilizing the sequential generation structure of diffusion models, the proposed model can "fine-tune" the inference hype-rparameters at each generation step, targeted to some black-bock function by only leveraging the function calls.

In addition, the proposed approach is proven to achieve some convergence rate for convex functions without smooth or strongly convex assumptions (though I haven't checked the proof and the correctness of the claim).

**Weaknesses:**

- The major weakness lies in the clarify of writing, which made it challenging for me to understand the exact mechanism of the method, and to check the claim on the convergence rate. In particular, (1) the indexing of $k$ and $t$ are confusing, and (2) which of the algorithm 1 qnd 2 is the practical algorithm? How are they different?


- The empirical evaluation is limited and unconvincing. E.g. In Figure 2, the extended dataset (last row) seems to have higher CLIP score, but lower visual quality. (1) While the author briefly touched upon this point and leaves it to future work, this behavior definitely limit the practical use of the proposed method. And I think it is a concern that needs to be well addressed within the scope of this work in order for it to be well-received in the community. (2) Furthermore, , I have reservations regarding whether the proposed method has effectively demonstrated superior practical utility, given that visually the samples from competing methods also align with the generation goal and can have better visual quality despite a lower target score.

Please see my detailed comments expanding the above two points in the below question section.

**Questions:**

1. Page 2, section 2.2 third line: "a relax the problem" -> "a relaxation of the problem"

2. Page 2, last paragraph, "Evolution Strategies (ES)", the acronym has been defined and used before.

3. Equation 4: describe the indexing of $t$ and $t+1$.

4. How is the proposed method different from DDOM?

5.  Section 3 Notation and symbols: a lot of them are introduced without providing the context. For example, in "denote $\bar \mu_k$ and $\bar \Sigma_k$ as the ... for Gaussian distribution", what "Gaussian distribution" is this referring to? And in the following sentence "Denote $\bar \theta_k$ ... as the parameter for optimization", what "optimization"? And "at $t^th$ iteration in optimization", what do you mean by iteration? Is the iteration in optimization the same thing as the generation step in regular diffusion model sampling?

6. Section 3: define $d$ (the dimension of data?)

7. Section 3: what value does $k$ take in (e.g. from 1 to some $K$ , where $K$ is the total of sampling steps)? What does $k$ and $t$ mean practically in $\bar \theta_k^t$? Is one of them the generation step index, and the other the dimension index?

8. Eq 5: from my impression of equation 5, $k$ should index the time step in diffusion sampling. However, here $k$ starts from $0$ which corresponds to random noise, and ends at $K$ which corresponds to the final data sample. But equation (1) (and equation (3)) uses $t$ to denote step and suggests a "reverse-time" manner, which seems not consistent.

9. When first introduced the black-box target score function $F(\cdot)$ (in the line above equation (6)), describe its argument, as in is it defined for $x_t$ at all steps or just the final one?

10. Page 3 last sentence, should choosing $\{\theta_{k-1}, \theta_{k-1}, \cdots, \theta_1\}$ proceed in a reverse order?

11. Equation 9, I wonder if it is correct to factorize the distribution w.r.t. $\bar \epsilon_k$ across $k$? If I understand it correctly, $\bar \epsilon_{k+1}$ contains $\bar \epsilon_k$ as a sub-component.

12. How is Algorithm 1 and 2 different from each other?

13. It seems like Algorithm 2 is the one used in practice (e.g. see the summarizing part in Section 1). Then why is it proposed in the Convergence Analysis section instead of the Method section?

14. Algorithm 1: $K$ should be an input argument. It is better to replace $\hat \mu_\phi (x_k, k)$ with $\hat \mu_\phi$ unless the arguments are defined. In the last two lines of the algorithm, what is $t+1$?

15. Page 7 Section 5.1, what does the subscript $m$ in $f(x) = \sum_{m=1}^d ... x-m2^2$ mean? Is it suppose to index the dimension of $x$? It seems to encode a different meaning of subscript $k$ used before.

16. In both experiments, how do you choose the total number of steps, and how do you choose the fine-tune steps to present, e.g. tabel 1 (steps =1596, 2646). Are they chosen randomly?

17. Does Figure 2 shows independent draws?

18. While the proposed method (last two panels of Figure2) can achieve a higher target score value, it is not clear that the visual quality is outperforming others, especially we see a deterioration in the last row. I would suggest addressing this concern, also provide other metrics to summarize the performance.

18.2 Furthermore, considering that the target score is determined by the CLIP score, which measures the proximity to the prompt "a close-up of a man with long hair," it appears that all the samples presented in Figure 2 reasonably align with the objective of achieving this generation goal. In this context, I have reservations regarding whether the proposed method has effectively demonstrated superior practical utility, particularly considering the lower visual quality of the generated content.

---

> ### Author Response · Authors · 2023-11-22
>
> We sincerely appreciate the reviewer's constructive advice and detailed comments. Our detailed responses to the reviewer's questions are presented as follows:
>
>
> $\textbf{W1}$: “The major weakness lies in the clarity of writing. … In particular, (1) the indexing of k and t are confusing. and (2) which of the algorithm 1 and 2 is the practical algorithm? How are they different? ”
>
>
> "(1) the indexing of k and t are confusing."
>
> $\textbf{A(1)}$: Thanks for your suggestion. We have revised the paper accordingly.  The index $k$ denotes the sequential step of the diffusion sampling.  The index  $t$ denotes the number of iterations for parameter updates in the black-box optimization process (see Alg.1 and Alg.2).
> To clarify the confusion, we revised section 2.1 to avoid the double definition of $t$ in  Eq.(1) $\sim$ (2).
>
> "(2) which of the algorithm 1 and 2 is the practical algorithm? How are they different? ”
>
> $\textbf{A(2)}$: Algorithm 2 is a general framework for sequential black-box optimization with a flexible set of parameters. Algorithm 1 is a practical algorithm of our framework (Alg.2) with a particular parameter setting for diffusion model targeted sampling. In Algorithm 1, the constraint of $\boldsymbol{H}_k^t$ in our framework(Alg.2) is relaxed, and the step size depends on the SDE solver coefficient.  In addition, the black-box function at each sequential step in Alg.1 is a particular function that depends on the diffusion model prediction.
>
> $\textbf{W2}$:  Loss of visual quality issue.  "The generated images have higher CLIP scores but lower visual quality. "
>
> $\textbf{A2}$: We have solved this issue and included additional experiments.  Please check our overall response and Section A in Appendix of our revised paper.
>
>
>
> $\textbf{Q3}$: “Equation 4: describe the indexing of t ”.
>
> $\textbf{A3}$: The index $t$ denotes the number of iterations for parameter updates in the black-box optimization process.
>
> $\textbf{Q4}$: “How is the proposed method different from DDOM?”
>
> $\textbf{A4}$: Our method is a novel sequential black-box optimization method that adaptively updates the mean and covariance matrices of a series of  Gaussian distributions for candidate sampling.      We employ our sequential black-box optimization method to fine-tune the inference of a pre-trained diffusion model for targeted sampling.
>
> In contrast,  the DDOM employs a weighted sampling of the training samples to train a conditional diffusion model from scratch and utilizes the conditional diffusion model for a targeted generation.
>
> $\textbf{Q5-Q6}$:
>  “Section 3 Notation and symbols: … what "Gaussian distribution" is this referring to? define d (the dimension of data?)”
>
> $\textbf{A5-A6}$: Thanks for your suggestion. We have revised the paper accordingly. Please check our revision.
>
> $\textbf{Q7}$: “Section 3: what value does k take in ?  What does k and t mean in $\bar{\theta}_k^t$?”
>
> $\textbf{A7}$:  $k$ take values from 1 to $K$, where $K$ denotes the number of diffusion sampling step. Note that the diffusion sampling is in a sequential manner.    In $\bar{\theta}_k^t$, index
> $k$ denotes the $k^{th}$ sequential  sampling step, where $t$ denotes the $t^{th}$ parameter update iteration in black-box optimization process.
>
>
> $\textbf{Q8}$: "But equation (1) (and equation (3)) uses $t$
>  to denote step and suggests a "reverse-time" manner, which seems not consistent."
>
> $\textbf{A8}$: We have revised Eq.(1) to Eq.(3) to avoid the double definition of $t$ (See Sec.2.1 in the revised paper).  The symbol $t$ denotes the number of iterations in the black-box optimization process (See our Alg.1 and Alg.2).
>
>
> $\textbf{Q9-Q10}$: "When first introduced the black-box target score function
>  (in the line above equation (6)), describe its argument"
>
> $\textbf{A9-A10}$: Thanks for your suggestion. We have revised the paper accordingly.
>
> The remaining concerns of the reviewer will be addressed in the next comment.

---

> ### Author Response · Authors · 2023-11-22
>
> $\textbf{Q11}$: “Equation 9, I wonder if it is correct to factorize the distribution w.r.t. $\bar{\epsilon}_k$ across $k$”
>
> $\textbf{A11}$: Note that $\bar{\boldsymbol{\epsilon}}_k=[\boldsymbol{\epsilon}_1^\top,\cdots,\boldsymbol{\epsilon}_k^\top]^\top$ for $k \in \{1,\cdots,K\}$ ,
> from the linear property of the expectation operation,  we know that
>
> $ \mathbb{E}_{\bar{\boldsymbol{\epsilon}}_K \sim \mathcal{N} (\bar{\boldsymbol{\mu}}_K,\bar{\boldsymbol{\Sigma}}_K) } [ \sum _{k=1}^K f_k(  \bar{\boldsymbol{\epsilon}}_k ) ] = \sum _{k=1}^K \mathbb{E} _{\bar{\boldsymbol{\epsilon}}_K  \sim  \mathcal{N} (\bar{\boldsymbol{\mu}}_K,\bar{\boldsymbol{\Sigma}}_K)   } [ f_k(  \bar{\boldsymbol{\epsilon}}_k ) ] $.  Note that $f_k( \bar{\boldsymbol{\epsilon}}_k)$  depends on $\bar{\boldsymbol{\epsilon}}_k =[\boldsymbol{\epsilon}_1^\top,\cdots,\boldsymbol{\epsilon}_k^\top]^\top$,  but it does not depends on
> $[\boldsymbol{\epsilon} _{k+1}^\top,\cdots,\boldsymbol{\epsilon}_K^\top]^\top$.
> Then we know $  \sum _{k=1}^K \mathbb{E} _{\bar{\boldsymbol{\epsilon}}_K  \sim  \mathcal{N} (\bar{\boldsymbol{\mu}}_K,\bar{\boldsymbol{\Sigma}}_K)   } [ f_k(  \bar{\boldsymbol{\epsilon}}_k ) ] =  \sum _{k=1}^K \mathbb{E} _{\bar{\boldsymbol{\epsilon}}_k  \sim  \mathcal{N} (\bar{\boldsymbol{\mu}}_k,\bar{\boldsymbol{\Sigma}}_k)   } [ f_k(  \bar{\boldsymbol{\epsilon}}_k ) ]$. Thus, the Eq.(9) (Eq.(8) in the revised paper) holds.
>
> $\textbf{Q12-Q13}$: “How is Algorithm 1 and 2 different from each other?”
>
> $\textbf{A12-A13}$:  Algorithm 2 is a general framework for sequential black-box optimization with a flexible set of parameters. Algorithm 1 is a practical algorithm of our framework (Alg.2) with a particular parameter setting for diffusion model targeted sampling. In Algorithm 1, the constraint of $\boldsymbol{H}_k^t$ in our framework(Alg.2) is relaxed, and the step size depends on the SDE solver coefficient.  In addition, the black-box function at each sequential step in Alg.1 is a particular function that depends on the diffusion model prediction.  Our convergence analysis is conducted on the general framework Alg.2 without relying on diffusion model prediction.
>
>
>
> $\textbf{Q14}$: "Algorithm 1: K
>  should be an input argument."
>
> $\textbf{A14}$: Thanks for the suggestion. We have revised the paper accordingly.
>
>
> $\textbf{Q15}$: “Page 7 Section 5.1, what does the subscript m mean? Is it suppose to index the dimension of  data.  It seems to encode a different meaning of subscript k”
>
> $\textbf{A15}$:  $m$ denotes the index of the element in the input vector $\boldsymbol{x}$.  $k$ is the index of the step in the diffusion sampling process.
>
>
> $\textbf{Q16:}$ "In both experiments, how do you choose the total number of steps, and how do you choose the fine-tune steps to present, e.g. tabel 1 (steps =1596, 2646). Are they chosen randomly?"
>
> $\textbf{A16:}$ Yes. The parameter is chosen randomly. We have reported additional experimental results in the Appendix of our revised paper.
>
> $\textbf{Q17:}$ “Does Figure 2 show independent draws?”
>
> $\textbf{A17:}$ Yes. The images are drawn independently. We report more comparisons and demonstrations in our revised paper.
>
> $\textbf{Q18:}$ "Loss of visual quality issue."
>
> $\textbf{A18:}$ We have solved the loss of visual quality issue and included additional experiments. Please check our overall response and Section A in the Appendix of our revised paper.

---

### Official Review · Reviewer_5v9t · 2023-11-01

**Soundness:** 3 good
**Presentation:** 3 good
**Contribution:** 3 good
**Rating:** 6
**Confidence:** 2

**Summary:**

This is the second paper on diffusion for black-box optimization. The previous work, DDOM, trains a conditional diffusion model using a reweighted objective function, and samples conditioning on the maximum y in the dataset and use classifier-free guidance.

The paper proposes to optimize an approximation by first order Taylor expansion and derives a closed-form update formula for the mean and variance of the score, computed by MC sampling. The convergence rate is proven

**Strengths:**

This paper offers an a method alternative to DDOM for diffusion-based black-box optimization. The method is well motivated and the performance is promising.

**Weaknesses:**

- What do you mean by a "full matrix update" and an "adaptive update"? When making claims such as "algorithm is the first full matrix adaptive black-box optimization with x convergence rate," it is important to explain the qualifiers and the relevant literature.
 - Can the proposed method be adapted to the tasks studied in DDOM and if so, how do they compare?
 - Page 4 Talor -> Taylor
 - Page 8 larger the score -> higher

**Questions:**

See weakness

---

> ### Author Response · Authors · 2023-11-22
>
> We sincerely appreciate the reviewer's constructive advice and valuable comments. Our detailed responses to the reviewer's questions are presented as follows:
>
> $\textbf{Q1}$: What do you mean by a "full matrix update" and an "adaptive update"? When making claims such as "algorithm is the first full matrix adaptive black-box optimization with x convergence rate," it is important to explain the qualifiers and the relevant literature.
>
>
> $\textbf{A1}$: Our algorithm updates the full covariance matrix at each step during the optimization process. The “adaptive update” means that the covariance matrix is updated according to the function values of the quires at each step instead of a fixed one.
>
>
> In the literature, NES,  CMAES and INGO are methods that update the full covariance matrix for black-box optimization, which has been discussed in our related work section.  In the NES and CMAES papers,  the authors did not provide convergence analysis.  In INGO, the convergence rate is analyzed for standard black-box problems under strongly convex assumptions.  Our convergence rate holds for sequential black-box optimization problems without strongly convex assumptions.  In particular, when the total step of the sequential problems is one, i.e., K=1.  We achieve a convergence rate for standard black-box problems.
>
>
>
>
> $\textbf{Q2}$: “Can the proposed method be adapted to the tasks studied in DDOM and if so, how do they compare?”
>
>
> $\textbf{A2}$: Thanks for your suggestion. We include additional experiments on tasks studied in DDOM. Please check our overall response and Section A in the Appendix of our revised paper.

---

### Author Response · Authors · 2023-11-22
**Overall Response**

We thank all reviewers’ constructive advice and valuable comments and appreciate the great efforts made by all reviewers, ACs, SACs and PCs.  We first present an overall response regarding the common concerns raised. Subsequently, we provide detailed responses to each reviewer’s specific questions.

We have revised the paper according to all reviewers’ suggestions and included additional experiments in Section A in the Appendix of our revised paper.

We want to highlight the following important updates and clarifications suggested by the reviewers.

$\textbf{Improve the visual quality of  the targeted generation}$

We revise the sequential black-box  objective $\sum_{k=1}^K f_k( \bar {\boldsymbol{\epsilon} }_k) $.  In our previous experiments, we set $f_k(\bar{\boldsymbol{\epsilon}}_k) = F(\boldsymbol{x}_k)$ for $k \in \{1,\cdots, K\}$, which call CLIP to evaluate the input (noised) image $\boldsymbol{x}_k$ at each diffusion sampling step $k$. This scheme is not effective enough because we care more about the generated image at the last step, i.e., $\boldsymbol{x}_K$,
than the generated image at the inner steps.

Thus,  we set the black-box function at the inner step as
$f_k(\bar{\boldsymbol{\epsilon}}_k) = F_k(\boldsymbol{x}_k)$ for $k \in \{1,\cdots, K-1\}$.
The black-box function $F_k(\boldsymbol{x}_k)$ takes the noised image $\boldsymbol{x}_k$ at the inner step as input and performs a deterministic sampling process for the future steps to achieve a predicted $\boldsymbol{x}_K$, and call CLIP to evaluate the predicted $\boldsymbol{x}_K$.

We evaluate the above update of our method on two targeted image generation cases: (1) "long hair man" (2)"Asian face".  The "Asian face"  is rare in the dataset. As a result, the targeted generation focuses more on out-of-distribution generation,  which is more challenging than the "long hair man" case.  For the "long hair man" case, we keep the target text as "a close-up of a man with long hair", which is the same as in our previous submission version. For the "Asian face" case, we set the target text as "a high quality close up of an asian".

For both cases, we set the number of iterations $T$ of our method as $T=120$.  The experimental results reported are at $T=120$. For all the methods in comparison, we employ the same backbone diffusion model with diffusion sampling step $K=14$.  We perform independent draws to generate $3,000$ images for evaluation.  The same set consists of $3,000$ i.i.d. sampled initial noise $\boldsymbol{x}_0\sim\mathcal{N}(0,\boldsymbol{I})$ is employed for all the methods to generate images for comparison.

The generated images are shown in Section A in the Appendix of our revised paper.
We can see that the visual quality of our method's generated images is better than DDOM's on the "Asian face" cases.  DDOM employs a reweighed sampling of training samples to train a conditional diffusion model from scratch.  This training scheme focuses on the tail of the distribution, which is more vulnerable to overfitting, especially for out-of-distribution target generation cases where the target image is rare in the training set.

$\textbf{Additional Comparison on the design tasks studied in DDOM paper}$

We further evaluate our method on the Superconductor and ChEMBL design tasks studied in the DDOM paper. The scores of different methods are reported in Table 4 in Section A in the Appendix of our revised paper.  Our method outperforms the DDOM on both the Superconductor and ChEMBL tasks, which shows the potential of our method on different domains beyond the targeted image generation.


$\textbf{Challenges on targeted image generation task studied in our paper}$

The black-box targeted image generation task is challenging due to the very high dimension of images. Specifically,   the dimension of the images employed in our work is $ \textbf{256} \times \textbf{256} \times \textbf{3} = \textbf{196608}$.  In contrast, most of the tasks employed in the DDOM paper are less than $100$-dimension, and the highest dimension is $5152$. The image generation task is more challenging than the one studied in DDOM paper.

---

### Meta-Review · Area_Chair_TG5C · 2023-12-05

**Metareview:**

The paper proposes a methodologically principled mechanism for conducting black-box optimization with diffusion models. In contrast to the prior work, the authors propose a novel fine-tuning strategy for the inference phase as a black-box sequential optimization problem. Overall the method is sound and novel. Unfortunately, as most reviewers pointed out the presentation of the idea is not reader-friendly to a general audience of researchers as the technique is over-formalized. In addition, the experimental section is weak, and real-world results are only shown on the CLIP score experiment using 30K images from the Celeb-HQ dataset. The AC believes the paper has a lot of merits and is technically strong, however, the clarity of writing and the breadth of the experiments need consideration for the next submission.

**Justification For Why Not Higher Score:**

Writing clarity and breadth of experiments are insufficient.

**Justification For Why Not Lower Score:**

N/A

---

### Decision · Program_Chairs · 2024-01-16

Reject